# The role of APOBEC3B in lung tumor evolution and targeted cancer therapy resistance

In this study, the impact of the apolipoprotein B mRNA-editing catalytic subunit-like (APOBEC) enzyme APOBEC3B (A3B) on epidermal growth factor receptor (EGFR)-driven lung cancer was assessed. *A3B* expression in EGFR mutant (EGFRmut) non-small-cell lung cancer (NSCLC) mouse models constrained tumorigenesis, while *A3B* expression in tumors treated with EGFR-targeted cancer therapy was associated with treatment resistance. Analyses of human NSCLC models treated with EGFR-targeted therapy showed upregulation of A3B and revealed therapy-induced activation of nuclear factor kappa B (NF-κB) as an inducer of *A3B* expression. Significantly reduced viability was observed with A3B deficiency, and A3B was required for the enrichment of APOBEC mutation signatures, in targeted therapy-treated human NSCLC preclinical models. Upregulation of *A3B* was confirmed in patients with NSCLC treated with EGFR-targeted therapy. This study uncovers the multifaceted roles of A3B in NSCLC and identifies A3B as a potential target for more durable responses to targeted cancer therapy.

Apolipoprotein B mRNA-editing catalytic subunit-like (APOBEC) enzymes are cytosine deaminases that have an important role in intrinsic responses to viral infection through deamination of deoxycytidine residues in viral single-stranded DNA[1,2]. APOBEC3 (A3) enzymes can act as potent host genome mutagens in multiple cancer types including non-small-cell lung cancer (NSCLC)[3,4]. In patients, both APOBEC3A (A3A)[5] and A3B[6] have been implicated to have a major role in NSCLC[3]. Earlier tumor genome sequencing studies revealed subclonal enrichment for mutations in an APOBEC substrate context, suggesting a possible role for this enzyme family in the acquisition of mutations later in tumor evolution[7–10]. Analysis of APOBEC3 family gene expression across multiple stages of lung adenocarcinoma revealed significantly elevated expression of *A3B* at multiple timepoints (adenocarcinoma in situ and invasive lung adenocarcinoma) compared to normal tissue[4].

While mouse models have contributed to our understanding of cancer evolution and drug responses[11–14], they lack the mutational heterogeneity observed in human tumors[15–17]. This may be due in part to the fact that mice encode only a single, cytoplasmic and nongenotoxic

APOBEC3 enzyme[18,19]. To understand the role of A3B in tumor evolution and therapy resistance, several mouse strains incorporating a human *A3B* transgene were engineered to mimic clonal and subclonal induction of *A3B* in oncogene-driven NSCLC and human preclinical models and clinical specimens were studied.

## Results

### A3B restrains tumor initiation in an epidermal growth factor receptor mutant (EGFRmut) lung cancer mouse model

The role of A3B in tumor initiation was first investigated in a mouse strain combining a new loxP-STOP-loxP (LSL) inducible human *A3B* transgenic model (*Rosa26^LSL-A3Bi*)[20] with a Cre-inducible EGFR^L858R-driven lung cancer mouse model (*TetO-EGFR^L858R; Rosa26^LNL-tTA*) to generate *EA3B* (*TetO-EGFR^L858R; Rosa26^LNL-tTA/LSL-A3Bi*)[11,12,21] mice (Fig. 1a). The tumor number and total tumor volume per mouse at 3 months postinduction, and the fraction of mice with tumors was significantly lower in *EA3B* mice than in *E* (*TetO-EGFR^L858R; Rosa26^LNL-tTA*) control mice (Fig. 1b,c and Extended Data Fig. 1a). A significantly decreased number of EGFR^L858R+ cells per lung area was also observed in *EA3B* mice versus *E* control mice

✉e-mail: deborah.caswell@crick.ac.uk; trever.bivona@ucsf.edu

(Fig. 1d). The programmed cell death marker caspase-3 was significantly higher in tumor cells of *EA3B* mice compared with *E* mice (Fig. 1e,f).

We hypothesized that *A3B* expression at tumor initiation in *EA3B* mouse models might induce increased chromosomal instability (CIN), p53 pathway activation and tumor cell death based on previous work[4]. In our models, a significantly higher fraction of lagging chromosomes and chromatin bridges were observed in anaphase tumor cells of *EA3B* mice compared with *E* mice[4] (Fig. 1g). There was also a significant increase in p53 nuclear positivity in tumors of *EA3B* mice compared with *E* mice that was not present at later stages (Extended Data Fig. 1b–d). No difference was observed in proliferation (Ki67) or DNA damage (γH2AX; Extended Data Fig. 1e,f). To assess if APOBEC activity contributes to increased tumor cell death at initiation, an EGFR[L858R] mouse model combined with a catalytically inactive form of A3B (*E(CAG) A3B[E255A]*)[22,23] was generated (Fig. 1h). The decrease in EGFR[L858R+] cells at 3 months postinduction observed with wildtype (WT) A3B was no longer observed in the enzyme inactive A3B mouse model (*E(CAG) A3B[E255A]*) compared with *E* control mice (Fig. 1h–j), suggesting that the increase in tumor cell death with A3B expression is at least in part due to the enzymatic activity of A3B.

We hypothesized that A3B expression could drive increased tumor cell death through enhanced immune surveillance in response to increased A3B activity[24]. A significant increase in both CD4 and CD8 T cells in *EA3B* mice was observed at 3 months postinduction (Extended Data Fig. 1g–i). Transplantation of an *EPA3B* mouse tumor cell line into WT C57BL/6J or *EPA3B* C57BL/6J transgenic mice resulted in the growth of EGFR[L858R+] A3B[+] tumors in *EPA3B* C57BL6/J transgenic mice but not WT C57BL/6J mice (Extended Data Fig. 1j–m), suggesting a level of immune tolerance to both the *EGFR[L858R]* and *A3B* transgenes.

Tumors were induced in an EGFR[L858R] p53-deficient mouse model either with or without *A3B* (*EP* and *EPA3B*; Fig. 1k and Extended Data Fig. 1j). No difference in the number of tumors at 3 months postinduction (Fig. 1l) or in overall survival (Fig. 1m) was observed in *EP* versus *EPA3B* mice, suggesting that *A3B* expression is tolerated in a p53-deficient model of EGFR-driven lung cancer. Thus, p53 in this model limits the tolerance of cancer cells to A3B expression at tumor initiation.

Next, CIN was assessed in systemic treatment-naïve (TN) patients with lung adenocarcinoma from the TRACERx421 (Tx421) cohort, confirming and expanding on previous findings from Tx100 (ref. 4). Tracking NSCLC evolution through therapy (TRACERx) is a prospective multicenter cancer study designed to delineate tumor evolution from diagnosis and surgical resection to either cure or disease recurrence. Tx100 was the analysis of the first 100 patients enrolled[9], while Tx421 was the analysis of the first 421 patients enrolled[25]. We considered the following three orthogonal approaches to estimate the extent of CIN in tumors: chromosome missegregation errors captured during anaphase; the amount of somatic copy-number alteration (SCNA) intratumor heterogeneity (ITH) between tumor regions (SCNA ITH)[25] and expression-based 70-gene CIN signature (CIN70)[4,26]. We observed a significant correlation between all three measures of CIN and A3B expression in both a subset of EGFRmut patients with lung adenocarcinoma in the Tx421 dataset (Fig. 2a–d) and patients with lung adenocarcinoma in the Tx421 dataset (Fig. 2e–h). Focusing on the genomic data, we observed a significant correlation between SCNA ITH and mutations in an APOBEC context (TCN/TCW C>T/G; Fig. 2i). These data together suggest that the increased CIN observed with *A3B* expression in EGFRmut mouse models is reflected in human NSCLCs in the Tx421 dataset.

**Subclonal *A3B* inhibits tumorigenesis.** Analysis of TN patients in the Tx421 cohort revealed that APOBEC-mediated mutagenesis is enriched subclonally in EGFRmut disease (Fig. 2j,k) and the wider cohort[9]. Mice in which *A3B* expression could be temporally separated from EGFR[L858R] expression (*EA3Bi*), allowing for induction of A3B expression in a subset of tumor cells within the already proliferating EGFRmut tumor, were generated to mirror subclonal APOBEC induction and to assess if subclonal

*A3B* expression decreased tumor cell death observed at initiation[11,12,21,27] (Extended Data Fig. 2a). *EA3Bi* mice had significantly lower tumor nodules per lung section and tumor area per lung area compared with *E* control mice (Extended Data Fig. 2b,c) along with significantly higher survival (Extended Data Fig. 2d). These data suggest that subclonal *A3B* also inhibits tumor growth, confirming the phenotype previously observed when A3B was induced concomitantly with EGFR[L858R] (Fig. 1a). Both mouse models (Fig. 1a and Extended Data Fig. 2a) are p53 WT.

**A3B promotes tyrosine kinase inhibitor (TKI) resistance.** Next, the impact of *A3B* on tumor evolution with EGFR TKI therapy was examined. Subclonal expression of *A3B* in TKI-treated *EA3Bi* mice drove a significant increase in tumor grade, tumor nodules per lung section and tumor area per tissue area compared with TKI-treated *Ei* control mice (Fig. 3a–d). Heterogeneous A3B tumor positivity (Fig. 3e) and a significant increase in A3B positivity with TKI therapy compared to untreated *EA3Bi* mice were observed (Fig. 3f). In an additional experiment, tumor growth and progression with TKI treatment were associated with a significant increase in tumor nodules and a substantial increase in tumor grade in *EA3Bi* mice compared with *Ei* control mice (Fig. 3g–i). Based on previous work illustrating an important role for uracil DNA glycosylase (UNG) in repairing APOBEC-induced uracil lesions[28], we evaluated UNG expression in A3B-expressing *EA3Bi* tumors. Staining for UNG revealed a significant decrease in UNG-positive cells per tumor in *EA3Bi* mice compared with *Ei* mice treated with TKI therapy (Fig. 3j,k). Taken together, these findings suggest that subclonal *A3B* expression with TKI therapy in conjunction with *UNG* downregulation contributes to increased tumor growth and TKI resistance.

Next, whole-exome sequencing (WES) was performed on TN and matched TKI-resistant mouse tumor cell lines (Extended Data Fig. 3a,b and Supplementary Table 1). A significantly higher number of mutations, as well as mutations in an APOBEC context, were detected in TKI-resistant A3B-expressing EGFRmut tumor cell lines (*EPA3B*) compared with control TKI-resistant EGFRmut tumor cell lines (*EP*), and compared with both control (*EP*) and *A3B*-expressing TN EGFRmut tumor cell lines (Extended Data Fig. 3a,b). Two unique de novo putative loss-of-function mutations in the protein tyrosine phosphatase receptor type S (*Ptprs*) gene were identified in an APOBEC context (Extended Data Fig. 3c). Loss of PTPRS function through mutation or deletion has been shown to increase TKI resistance in multiple human preclinical cancer models and has been linked with worse overall survival and more rapid disease progression in patients with EGFR-driven lung cancer[29–31]. The equivalent of the A3B-driven mutation in humans (Ptprs_mut1, D138N; Extended Data Fig. 3c) was identified in tumors of patients with lung, colorectal and bladder cancer from The Cancer Genome Atlas (TCGA) and in one EGFR[L858R] TRACERx patient with NSCLC (Extended Data Fig. 3d).

To validate our findings from mouse models, long-term cell viability with targeted therapy was assessed in established human cell line models of oncogenic EGFRmut and echinoderm microtubule-associated protein-like 4-anaplastic lymphoma kinase (EML4-ALK) lung adenocarcinoma with CRISPR-mediated *A3B* depletion. Under EGFR TKI treatment (osimertinib), A3B-depleted PC9 and HCC827 lines (harboring EGFR[exon19del]; Extended Data Fig. 4a–d) showed significantly reduced cell viability compared to A3B-competent control lines (Fig. 3l,m). Similarly, a significant reduction in cell viability was observed in an *A3B*-knockout (KO) EML4-ALK cancer cell line (H3122; Extended Data Fig. 4e,f) treated with the Food and Drug Administration-approved ALK TKI alectinib (Fig. 3n). KO of *A3B* had no effect on cell viability in untreated PC9, HCC827 or H3122 cell lines (Extended Data Fig. 4g–i). These data suggest that *A3B* expression confers enhanced cell survival with targeted therapy.

**Targeted therapy induces A3B expression and UNG downregulation.** Our mouse lung cancer models demonstrated that *A3B* expression is associated with targeted therapy resistance. We hypothesized that

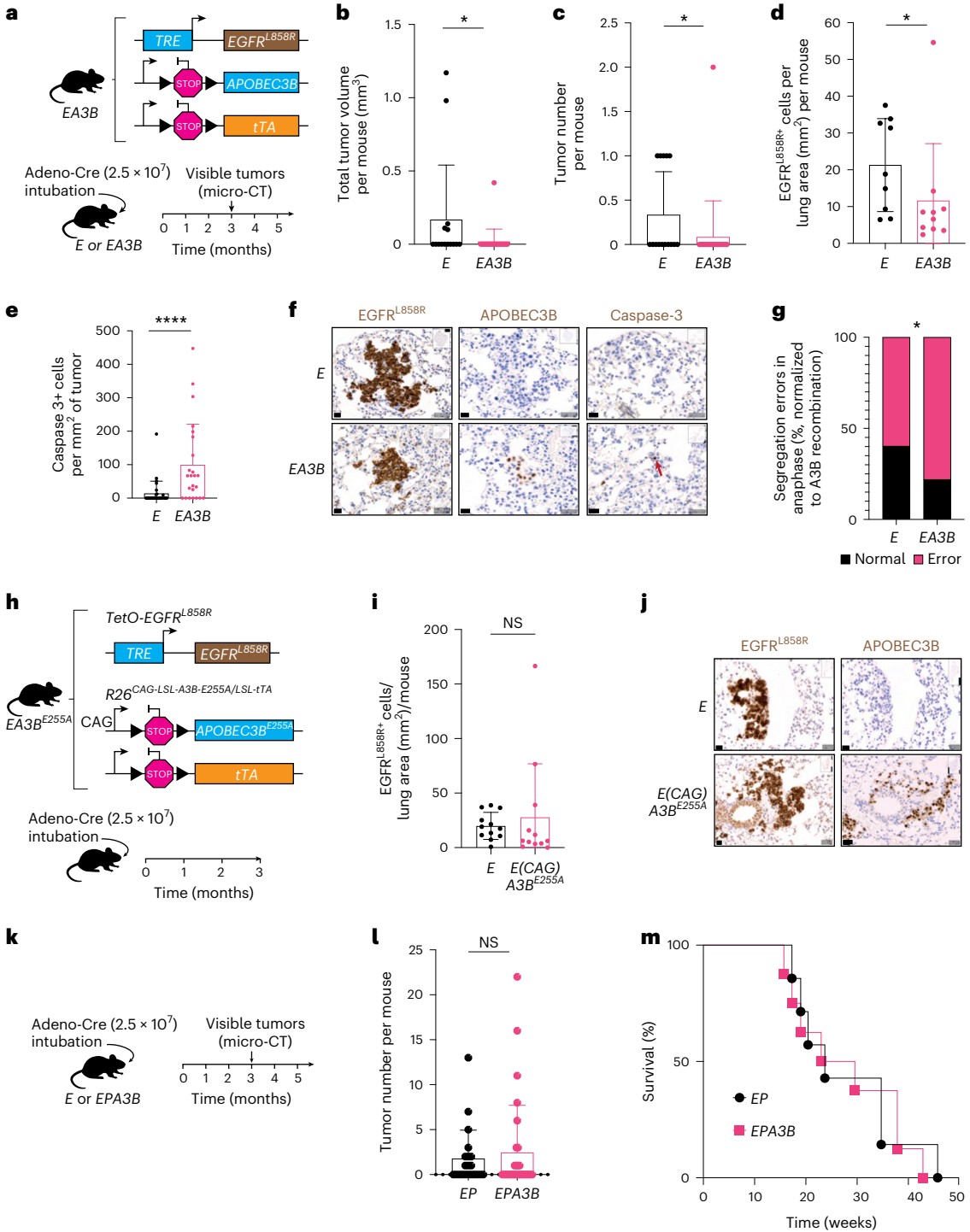

**Fig. 1 | Continuous *APOBEC3B* expression is detrimental for tumorigenesis in a p53 WT EGFR[L858R] mouse model of lung cancer. a**, Tumorigenesis in *E* (*TetO-EGFR[L858R];Rosa26[LNL-tTA]*) and *EA3B* (*TetO-EGFR[L858R];Rosa26[LNL-tTA/LSL-A3Bi]*) mice was induced using the indicated viral titer. Tumor growth was assessed by micro-CT analysis. **b**, Total tumor volume per mouse at 3 months postinduction quantified by micro-CT analysis (*E*, *n* = 15; *EA3B*, *n* = 24; mean ± s.d., two-sided Mann–Whitney test, *\*P* = 0.0163, each dot represents a mouse). **c**, Total tumor number per mouse at 3 months postinduction quantified by micro-CT analysis (*E*, *n* = 15; *EA3B*, *n* = 24, mean ± s.d., two-sided Mann–Whitney test, *\*P* = 0.0236, each dot represents a mouse). **d**, Quantification of EGFR[L858R+] cells per lung area (mm²) by IHC staining at 3 months postinduction (*E*, *n* = 9; *EA3B*, *n* = 10; mean ± s.d., two-sided Mann–Whitney test, *\*P* = 0.0435, each dot represents a mouse). **e**, Quantification of caspase 3+ cells per mm² of tumor at 3 months postinduction (*E*, *n* = 9; *EA3B*, *n* = 10; mean ± s.d., two-sided Mann–Whitney test,

\*\*\*\**P* < 0.0001, each dot represents a tumor). **f**, Representative IHC stainings of EGFR[L858R], APOBEC3B and caspase-3 (scale bar = 20 µm, arrow indicates positive cell; *E*, *n* = 9; *EA3B*, *n* = 10 biological replicates). **g**, Percent chromosome missegregation errors at 3 months postinduction (two-sided Fisher's exact test, *\*P* = 0.016; *E*, *n* = 9; *EA3B*, *n* = 10). **h**, Tumorigenesis in *E* and *E(CAG)A3B[E255A]* mice was induced using the indicated viral titer (2.5 × 10⁷ viral particles per mouse). **i**, Quantification of EGFR[L858R+] cells per lung area (mm²) by IHC staining at 3 months postinduction (*E*, *n* = 12; *E(CAG)A3B[E255A]*, *n* = 12; mean ± s.d., each dot represents a mouse). **j**, Representative IHC staining of EGFR[L858R] and APOBEC3B (scale bar = 20 µm; *E*, *n* = 12; *E(CAG)A3B[E255A]*, *n* = 12). **k**, Tumor growth was assessed by micro-CT analysis in *EP* and *EPA3B* mice. **l**, Total tumor number per mouse at 3 months postinduction quantified by micro-CT analysis (*EP*, *n* = 21; *EPA3B*, *n* = 30; combined from two separate experiments). **m**, Survival curve of *EP* versus *EPA3B* mice (*EP*, *n* = 8; *EPA3B*, *n* = 7; each dot represents a mouse). NS, not significant.

targeted therapy may induce adaptations that increase the expression of A3 family members and decrease the expression of *UNG* in human models. Based on current literature[4,5,32,33], mRNA expression levels of *A3A*, *A3B*, *APOBEC3C* (*A3C*) and *APOBEC3F* (*A3F*) were measured. In PC9 cells, a significant increase in all four members was observed with osimertinib, with *A3A* being the most significantly elevated (Fig. 4a). In HCC827 cells, *A3A* and *A3B* were the most significantly elevated, with both induced to similar levels with osimertinib (Fig. 4b). A significant increase in overall APOBEC activity (Fig. 4c,d) and A3B protein levels (Fig. 4e,f) were also observed. Each *A3* gene was then silenced using small interfering RNAs (siRNAs) specific for each family member (Extended Data Fig. 5a–i), and APOBEC activity was assessed. Only knockdown of *A3B* resulted in a significant decrease in APOBEC activity with TKI therapy in PC9 and HCC827 cell lines (Fig. 4g). These data suggest that while several A3 family members likely contribute to the increased APOBEC activity observed with TKI therapy, A3B appears to be a major contributor.

Targeted therapy-induced transcriptional changes of *A3B* and *UNG* were assessed in established human lung cancer cell line data from publicly available datasets (Gene Expression Omnibus (GEO) database, GEO2R). Treatment of EGFRmut cell lines (HCC827, PC9 and HCC4006 harboring EGFR[L747-E749del,A750P]) with the EGFR TKI erlotinib was associated with transcriptional upregulation of *A3B* both acutely (6-h to 1-d treatment) and at later timepoints (8-d treatment; Fig. 5a). These transcriptional changes were confirmed in an independent RNA-seq (RNA sequencing) dataset[34] with a significant upregulation of *A3B* and downregulation of *UNG* following osimertinib treatment (Fig. 5b), suggesting a conserved effect of EGFR pharmacologic inhibition independent of the generation (evolution of targeted therapy development leading to more specific and effective molecules) of EGFR inhibitor.

Transcriptional upregulation of *A3B* and downregulation of *UNG* were subsequently validated in multiple oncogenic EGFR-driven cellular models of lung adenocarcinoma at both the RNA (Fig. 5c,d) and protein levels (Fig. 5e). To rule out off-target pharmacological effects of EGFR TKIs, *A3B* expression was examined with siRNA-mediated silencing of *EGFR* and also led to *A3B* upregulation and *UNG* downregulation (Fig. 5f). Induction of *A3B* was also observed upon treatment with an inhibitor of mitogen-activated protein kinase kinase (MAP2K or MEK1) (selumetinib; Fig. 5a). The induction of *A3B* by different inhibitors of oncogenic receptor tyrosine kinases (RTKs) and their downstream signaling components, such as MEK1, indicates that upregulation of A3B is likely a consequence of oncogenic signaling inhibition, and not specific to EGFR TKIs.

Consistent with RNA and protein level changes, TKI treatment resulted in a significant increase in nuclear APOBEC activity[35] and decrease in nuclear uracil excision capacity of UNG in multiple EGFR-driven cell line models, including EGFR[exon19del] cells (PC9 and HCC827) and EGFR[L858R+T790M] cells (H1975; Fig. 4c,d and Extended Data Fig. 6a–e). Increased *A3B* expression and APOBEC activity as well as decreased *UNG* expression and uracil excision activity were also observed in EML4-ALK-driven cellular models (H3122 and H2228) during ALK TKI treatment (Extended Data Fig. 6f–i).

*A3B* was then stably knocked down using small hairpin RNA (shRNA) in PC9 cells, and rescue experiments with expression vectors containing either WT A3B tagged with human influenza hemagglutinin (HA) (A3B WT-HA tagged) or catalytically inactive A3B tagged with HA (A3B E225A-HA tagged) were performed. APOBEC activity with A3B knockdown was significantly reduced with TKI treatment versus A3B-proficient lines with TKI treatment (Extended Data Fig. 6j). Expression of the WT catalytically active, but not the mutant catalytically inactive A3B, rescued the decline in nuclear APOBEC activity caused by A3B depletion (Extended Data Fig. 6j–l). While knockdown of *A3B* induced no off-target reductions in any other *A3* family members, significant increases in *A3A*, *A3G* and *A3H* expression were detected (Extended Data Fig. 6m), corroborating previous reports in human breast and lymphoma cancer cell lines showing increased *A3A* expression with A3B loss[36]. These data suggest that A3B is a substantial contributor to the increased APOBEC activity observed with TKI treatment.

To exclude an indirect effect of targeted therapy on cell cycle arrest that might alter APOBEC enzyme expression, EGFRmut NSCLC PC9 cells were treated with the CDK4/6 cell cycle inhibitor palbociclib[37]. Palbociclib treatment-induced G0/G1 cell cycle arrest with a comparable arrest measured with osimertinib (Extended Data Fig. 6n). *UNG* expression decreased upon palbociclib treatment; however, there was a significant decline in *A3B* expression (Extended Data Fig. 6o), contrasting with the increased expression observed upon TKI therapy and suggesting that TKI-mediated induction of *A3B* is unlikely to be a consequence of TKI treatment-induced cell cycle inhibition.

*A3B* and *UNG* expression levels were then examined in multiple human tumor xenograft models. An increase in A3B and a decrease in UNG protein levels were detected in EGFR TKI-treated tumor tissues from three distinct oncogenic EGFR-driven CDX models of human lung adenocarcinoma (Extended Data Fig. 7a–f). Additionally, RNA-seq analyses from an EGFR[L858R]-harboring patient-derived xenograft (PDX) model of lung adenocarcinoma[38] revealed a nonsignificant increase in *A3B* mRNA and a decrease in *UNG* mRNA levels upon treatment with erlotinib (Extended Data Fig. 7g), and significant increase in A3B and a nonsignificant decrease in UNG with osimertinib[34] (Extended Data Fig. 7h). These findings support a model whereby EGFR oncoprotein inhibition induces increased *A3B* expression and decreased *UNG* expression.

**Fig. 2 | *APOBEC3B* expression correlates with multiple measures of CIN, and APOBEC mutagenesis is subclonally enriched in TN EGFRmut patients from the TRACERx421 (Tx421) dataset. a**, Correlation between *APOBEC3B* (*A3B*) expression and percent missegregation errors calculated using patients with EGFRmut lung adenocarcinoma (n = 13 tumors; Spearman, R = 0.59; P = 0.038). **b**, Significant correlation between *A3B* expression and CIN70 GSEA score calculated using EGFRmut tumors from patients with lung adenocarcinoma (n = 19 tumors; Spearman, R = 0.59; P = 0.009). **c**, Significant correlation between *A3B* expression and CIN70 GSEA score calculated using EGFRmut tumor regions in patients with lung adenocarcinoma (n = 42 tumor regions; Spearman, R = 0.64; P < 9 × 10⁻⁶). **d**, Correlation between *A3B* expression and subclonal CIN fraction calculated in EGFRmut patients with lung adenocarcinoma (n = 19 tumors; bootstrapped Spearman, R = 0.5; P = 0.032). **e**, Significant correlation between percent missegregation errors (anaphase bridges (bridges) and lagging chromosomes (lagging)) and CIN70 score calculated using tumors from patients (n = 112 tumors; Spearman, R = 0.27; P = 0.0038). **f**, Significant correlation between *A3B* expression and CIN70 GSEA score calculated using tumors from patients with lung adenocarcinoma (n = 188 tumors; Spearman, R = 0.56; P < 2 × 10⁻¹⁶). **g**, Significant correlation between *A3B* expression and CIN70 GSEA score calculated using tumor regions in patients with lung adenocarcinoma (n = 466 tumor regions; Spearman, R = 0.54; P < 2 × 10⁻¹⁶). **h**, Correlation between *A3B* expression and subclonal CIN fraction calculated patients with lung adenocarcinoma in the Tx421 cohort (n = 168 tumors; bootstrapped Spearman, R = 0.26; P = 0.00087). **i**, Comparisons between C>T and C>G mutation counts at TCN and TCW trinucleotide context and percentage of genome altered subclonally (n = 25, two-sided Pearson, TCW R = 0.49, P = 0.015; TCN R = 0.52, P = 0.0092). **j**, Comparison of clonal and subclonal APOBEC-associated mutation signature (clonal APOBEC–subclonal APOBEC) in patients with EGFR driver mutations (l, la, exon 19 deletion). White bars indicate that the patient is *TP53* WT or has a subclonal *TP53* mutation. Red bars indicate that the patient has a clonal *TP53* mutation (n = 23, one-sided Wilcoxon, P = 1 × 10⁻⁴). **k**, Number of APOBEC-associated mutations in patients with EGFR driver mutations (l, la, exon 19 deletion). Colors indicate clonal or subclonal APOBEC or non-APOBEC-associated mutations (n = 23). All analyses were performed on samples from the Tx421 cohort. GSEA, gene set enrichment analysis; NES, normalized enrichment score; TMM, trimmed mean of *M* values.

## Nuclear factor-kappa B (NF-κB) signaling contributes to TKI-induced A3B upregulation

Prior work from our group and others revealed that NF-κB signaling is activated upon EGFR oncogene inhibition in human lung cancer as a stress and survival response[38]. Previous data suggest that NF-κB signaling may be a prominent inducer of *A3B* gene expression[39,40]. We hypothesized that NF-κB signaling activation upon targeted therapy promotes *A3B* upregulation. To test this hypothesis, an established RNA-seq dataset generated from EGFR-driven human lung adenocarcinoma cells treated acutely with either erlotinib or an NF-κB

inhibitor (PBS-1086) or both in combination was examined[38]. TKI treatment-induced transcriptional upregulation of *A3B* was attenuated by cotreatment with the NF-κB inhibitor[38] (Extended Data Fig. 8a), suggesting that the NF-κB pathway induces *A3B* expression. To confirm this, the NF-κB pathway was activated with increasing concentrations of Tumor necrosis factor-α, which elevated nuclear RELA and RELB as well as nuclear A3B protein levels (Extended Data Fig. 8b) and cellular *A3B* mRNA expression (Extended Data Fig. 8c). Inhibition of the NF-κB pathway by simultaneous depletion of both *RELA* and *RELB* (Extended Data Fig. 8d) reduced TKI-induced *A3B* mRNA expression (Extended Data

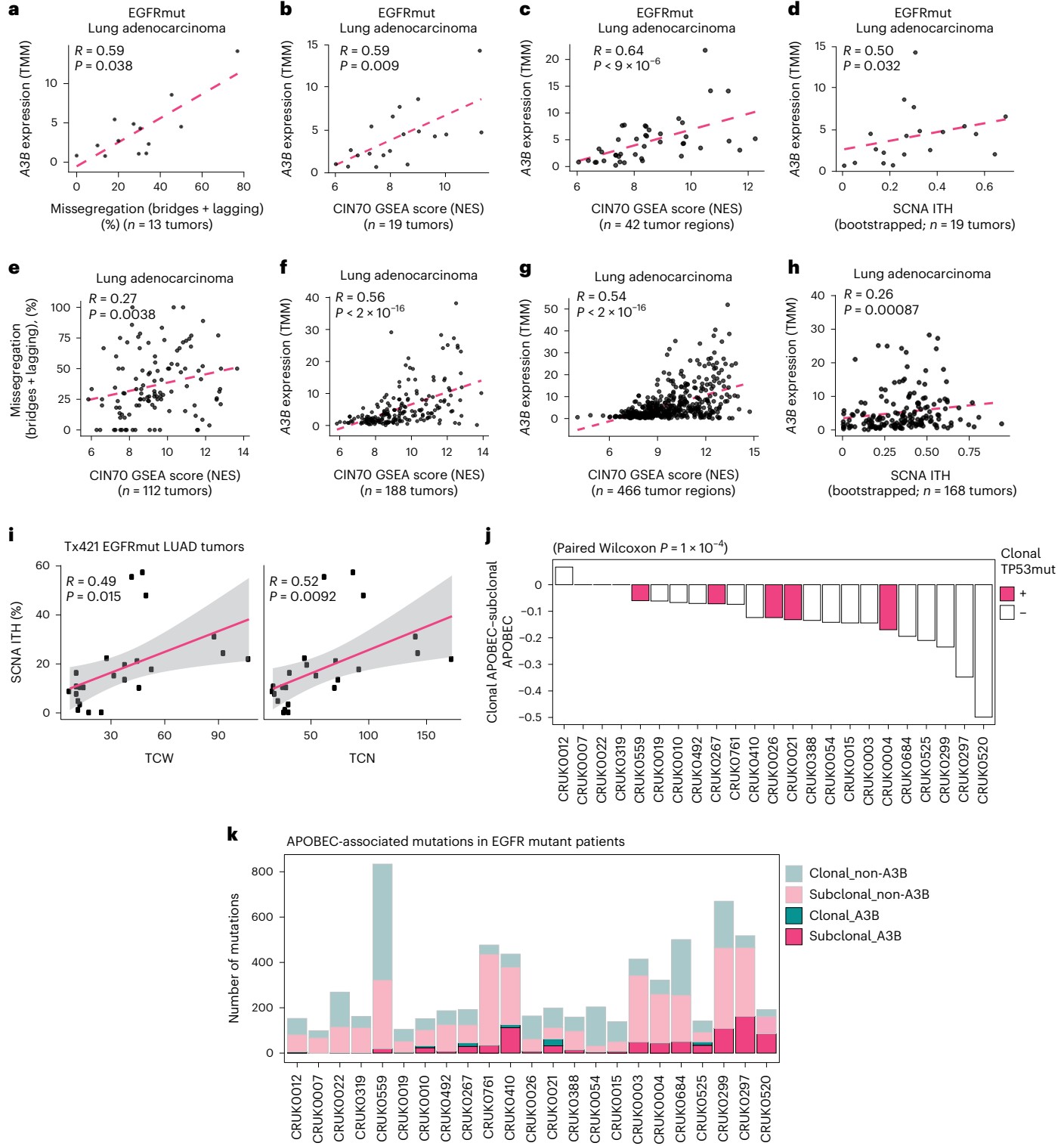

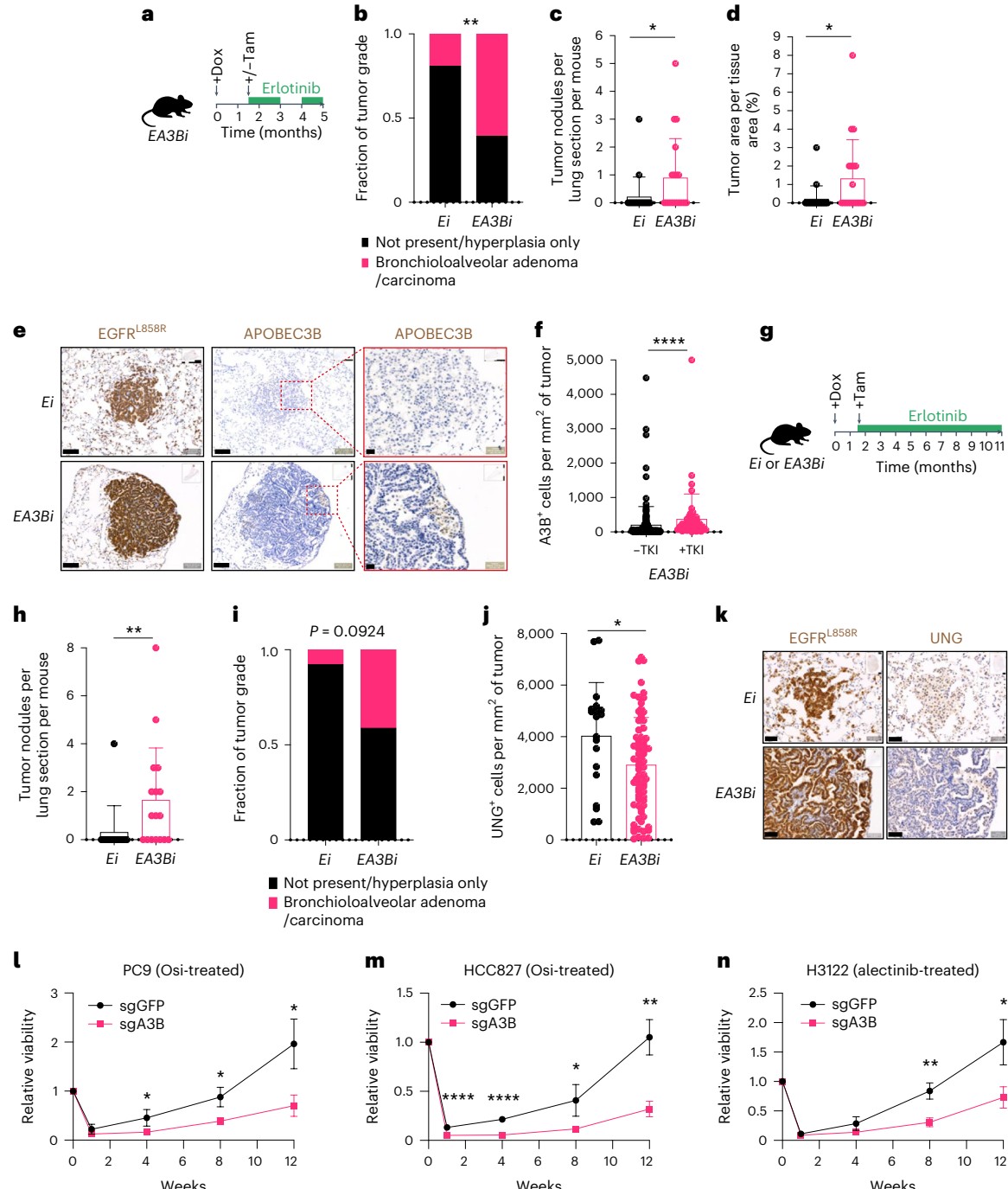

**Fig. 3 | APOBEC3B drives targeted therapy resistance in mouse and human preclinical models. a**, *TetO-EGFRL858R;CCSP-rtTA;R26LSL-APOBEC3B/Cre-ER(T2)* mice with or without induction of subclonal *APOBEC3B (A3B)* with TKI therapy (erlotinib). **b**, Fraction of tumor grade, not present or hyperplasia only. Bronchioloalveolar adenoma or carcinoma at 5 months (*Ei*, n = 19; *EA3Bi*, n = 19; two-sided Fisher's exact test, **P = 0.0044). **c**, Tumor nodules per lung section per mouse at 5 months (*Ei*, n = 19; *EA3Bi*, n = 19; two-sided Mann–Whitney test, *P = 0.0443). **d**, Tumor area per lung area per mouse at 5 months (*Ei*, n = 19; *EA3Bi*, n = 19; two-sided Mann–Whitney test, *P = 0.0212). **e**, Representative IHC staining of EGFR[L858R] and A3B (scale bar = 100 μm and 20 μm; *Ei*, n = 19; *EA3Bi*, n = 19 biological replicates). **f**, A3B[+] cells per mm² of tumor per mouse (*EA3Bi* − TKI = 151, *EA3Bi* +TKI = 52, two-sided Mann–Whitney test, ****P < 0.0001). **g**, Induction of subclonal A3B using *TetO-EGFRL858R;CCSP-rtTA;R26Cre-ER(T2)/+* or *TetO-EGFRL858R;CCSP-rtTA;R26LSL-APOBEC3B/Cre-ER(T2)* mice with

continuous TKI therapy (erlotinib). **h**, Tumor nodules per lung section per mouse (*Ei*, n = 13; *EA3Bi*, n = 17; two-sided Mann–Whitney test, **P = 0.0086). **i**, Fraction of tumor grade, not present or hyperplasia only. Bronchioloalveolar adenoma or carcinoma at 11 months (*Ei*, n = 13; *EA3Bi*, n = 17; two-sided Fisher's exact test). **j**, Quantification of UNG[+] cells per mm² of tumor at 5 months postinduction (*E*, n = 10; *EA3Bi*, n = 10; two-tailed *t* test, *P = 0.0226, each dot represents a tumor). **k**, Representative IHC staining of EGFR[L858R] and UNG. Scale bar = 50 μm. **l–n**, CellTiter-Glo viability timecourse assays performed on *A3B*-deficient or *A3B*-proficient PC9 cells treated with 100 nM Osi (**l**, n = 3 biological replicates, mean ± s.d., two-sided *t* test, *P = 0.0439, *P = 0.0155, *P = 0.0168); HCC827 cells treated with 100 nM Osi (**m**, n = 3 biological replicates, mean ± s.d., two-sided *t* test, *P = 0.0377, **P = 0.0029, ****P = 0.0004, ****P = 0.00009); H3122 cells treated with 100 nM alectinib (**n**, n = 3 biological replicates, mean ± s.d., two-sided *t* test, *P = 0.0189, **P = 0.0044). Osi, osimertinib.

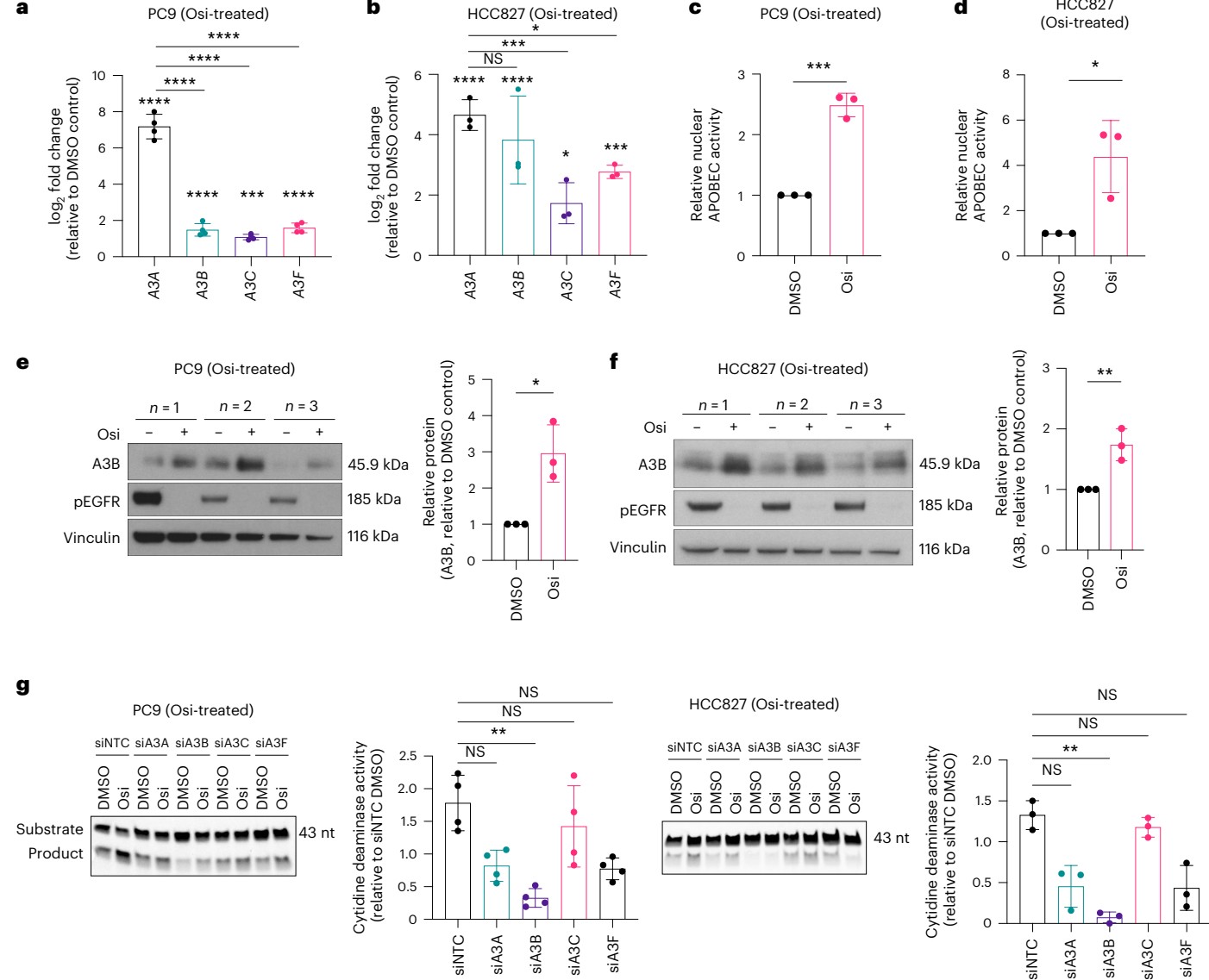

**Fig. 4 | Knockdown of *APOBEC3B* reduces the TKI therapy-induced APOBEC activity in EGFRmut lung cancer cell lines. a**, RT–qPCR performed on PC9 cells treated with DMSO or 0.5 μM Osi for 18 h, measuring *APOBEC3A* (*A3A*), *APOBEC3B* (*A3B*), *APOBEC3C* (*A3C*) and *APOBEC3F* (*A3F*; *n* = 4 biological replicates, mean ± s.d., one-way ANOVA test, ***P = 0.0002, ****P < 0.0001). **b**, RT–qPCR analysis of HCC827 cells treated with DMSO or 0.5 μM Osi for 18 h (*n* = 3 biological replicates, mean ± s.d., one-way ANOVA test, *P = 0.0264, ***P = 0.0005, ***P = 0.0008, ****P < 0.0001). **c**, APOBEC activity assay performed using nuclear extracts of PC9 cells treated with DMSO or 2 μM Osi for 18 h (*n* = 3 biological replicates, mean ± s.d., two-tailed *t* test, ***P = 0.0002). **d**, APOBEC activity assay using nuclear extracts of HCC827 cells treated with DMSO or 0.4 μM Osi for 18 h (*n* = 3 biological replicates, mean ± s.d., two-tailed *t* test, *P = 0.0213). **e**, Western blot analysis of A3B protein levels in PC9 cells treated with DMSO or 0.5 μM Osi for 18 h with quantification (*n* = 3 biological replicates, mean ± s.d., two-tailed *t* test, *P = 0.0129). **f**, Western blot analysis for A3B protein levels in HCC827 cells treated with DMSO or 0.5 μM Osi for 18 h (*n* = 3 biological replicates, mean ± s.d., two-tailed unpaired *t* test, **P = 0.0082). **g**, APOBEC activity assay performed on lysates of PC9 or HCC827 cells treated with DMSO or 0.5 μM Osi for 18 h, with siRNA knockdown of *APOBEC3A* (siA3A), *APOBEC3B* (siA3B), *APOBEC3C* (siA3C) and *APOBEC3F* (siA3F) and nontargeting siRNA (siNTC), and quantification (PC9, *n* = 4 biological replicates, mean ± s.d., one-way ANOVA test (nonparametric), **P = 0.0017; HCC827, *n* = 3 biological replicates, mean ± s.d., one-way ANOVA test (nonparametric), **P = 0.0076). ANOVA, analysis of variance.

Fig. 8e) and A3B protein levels (Extended Data Fig. 8f). Co-inhibition of EGFR and NF-κB pathways blocked EGFR inhibition-induced *A3B* upregulation in oncogenic EGFR-driven NSCLC xenografts (Extended Data Fig. 7c,d). Codepletion of both NF-κB transcription factors RELA and RELB impaired TKI-induced nuclear APOBEC activity (Extended Data Fig. 8g). These data support NF-κB activation with EGFR TKI treatment as an inducer of *A3B* upregulation in response to therapy.

To investigate the clinical relevance of these findings, we examined single-cell RNA-seq data in an established dataset obtained from clinical specimens of NSCLC procured from patients at the following three timepoints: (1) treatment naïve before initiation of systemic targeted therapy (classified as TN), (2) while on targeted therapy when the tumor was regressing or at stable state as evaluated by standard clinical imaging (classified as residual disease (RD)) and (3) at clear progressive disease (PD, acquired resistance) as determined by standard clinical imaging (classified as PD). The classification of response was based on Response Evaluation Criteria in Solid Tumors (RECIST) criteria[41]. In total, 66 samples obtained from 30 patients with lung cancer pre-TKI or post-TKI therapy (erlotinib (EGFR), osimertinib (EGFR) and crizotinib (ALK) being the most frequent targeted therapies) were

**a**

HCC827 6 h erlotinib vs. DMSO (GSE51212)

| Gene symbol | log(FC) | $P$ | $P_{adj.}$ |
|---|---|---|---|
| *APOBEC3B* | 1.97 | $5.37 \times 10^{-10}$ | $2.98 \times 10^{-6}$ |
| *UNG* | −0.656 | $5.67 \times 10^{-6}$ | 0.000337 |

HCC4006 1 d erlotinib vs. DMSO (GSE57156)

| Gene symbol | log(FC) | $P$ | $P_{adj.}$ |
|---|---|---|---|
| *APOBEC3B* | 1.29 | 0.000226 | 0.009784 |
| *UNG* | −0.907 | 0.000888 | 0.022115 |

PC9 8 d erlotinib vs. DMSO (GSE67051)

| Gene symbol | log(FC) | $P$ | $P_{adj.}$ |
|---|---|---|---|
| *APOBEC3B* | 0.494 | $7.64 \times 10^{-3}$ | 0.039366 |
| *UNG* | −0.204 | 0.0673 | 0.176893 |

HCC827 6 h selumetinib (AZD6244) vs. DMSO (GSE51212)

| Gene symbol | log(FC) | $P$ | $P_{adj.}$ |
|---|---|---|---|
| *APOBEC3B* | 0.572 | $1.08 \times 10^{-5}$ | $4.69 \times 10^{-3}$ |
| *UNG* | −0.135 | $5.73 \times 10^{-2}$ | $3.62 \times 10^{-1}$ |

**b**

PC9 9 d Osi vs. DMSO

| Gene symbol | log(FC) | $P$ | $P_{adj.}$ |
|---|---|---|---|
| *APOBEC3B* | 2.25 | $5.69 \times 10^{-71}$ | $4.28 \times 10^{-69}$ |
| *UNG* | −0.797 | $2.26 \times 10^{-09}$ | $1.68 \times 10^{-8}$ |

**Fig. 5 | Treatment with TKI induces *APOBEC3B* upregulation. a**, GSEA of the indicated GEO2R datasets of EGFR-driven cellular models of human lung adenocarcinoma treated with erlotinib or a mitogen-activated protein kinase kinase (MAP2K or MEK1) inhibitor (AZD6244). **b**, RNA-seq analysis of gene expression changes in PC9 cells treated with 2 μM Osi for 9 d relative to DMSO-treated cells ($n = 3$ biological replicates, mean ± s.d., ANOVA test). **c**, RT–qPCR analysis of PC9 cells treated with DMSO or 2 μM Osi for 18 h ($n = 4$ biological replicates, mean ± s.d., one-way ANOVA test, *$P = 0.0349$, ****$P < 0.0001$). **d**, RT–qPCR analysis of HCC827 cells treated with DMSO or 0.4 μM osimertinib for 18 h ($n = 4$ biological replicates, mean ± s.d., one-way ANOVA test, ***$P = 0.0008$, **$P = 0.0014$). **e**, Western blot analysis of cells treated in **a** and **b** (CYTO, cytoplasmic extracts; H3, histone H3; NUC, nuclear extracts) with quantification of A3B levels in PC9 cells ($n = 3$ biological replicates, mean ± s.d., one-way ANOVA test, **$P = 0.0012$, **$P = 0.0058$) and HCC827 cells ($n = 3$ biological replicates, mean ± s.d., one-way ANOVA test, *$P = 0.0186$). **f**, RT–qPCR analysis of PC9 cells treated with nontargeting siRNA (siNTC) or *EGFR* siRNA (siEGFR) for 18 h and grown for 2 d ($n = 4$ biological replicates, mean ± s.d., two-sided $t$ test, **$P = 0.0075$, ***$P = 0.0002$, **$P = 0.0027$). FC, fold change.

analyzed (Supplementary Table 2a). We observed that mRNA expression of *A3B* and NF-κB components *RELA* and *RELB*, as well as an NF-κB gene signature[42], were significantly increased in tumors exposed to EGFR TKI treatment, in particular at tumor progression with therapy (Extended Data Fig. 8h–k).

### *UNG* downregulation is associated with *c-JUN* suppression during TKI treatment

We next investigated the mechanism of *UNG* downregulation during targeted therapy. *UNG* gene promoter analysis (using PROMO)[43] revealed the presence of predicted JUN consensus binding sites. RNA-seq data from EGFR TKI-treated PC9 cells indicated that like *UNG*, *c-JUN* was also transcriptionally downregulated upon treatment, which was validated using RT–qPCR (Extended Data Fig. 8l). This aligns with the expected downregulation of *c-JUN* upon inhibition of the mitogen-activated

protein kinase (MAPK) pathway during EGFR inhibition by TKI treatment[44]. We hypothesized that TKI treatment-induced *UNG* downregulation could be caused by *c-JUN* downregulation. Silencing of *c-JUN* by siRNA was sufficient to suppress *UNG* expression, suggesting that *UNG* downregulation could be a consequence, in part, of the *c-JUN* suppression that occurs during TKI-mediated MAPK signaling suppression (Extended Data Fig. 8m).

### A3B is required for APOBEC mutation signature accumulation during targeted therapy

To examine the role of A3B expression on mutagenesis during targeted therapy, *A3B*-deficient and *A3B*-proficient single-cell cloned PC9 cells (Extended Data Fig. 4a,b) were treated with osimertinib using a dose-escalation protocol to resistance (3 months; Fig. 6a). The mutations and proportion of APOBEC mutation signatures

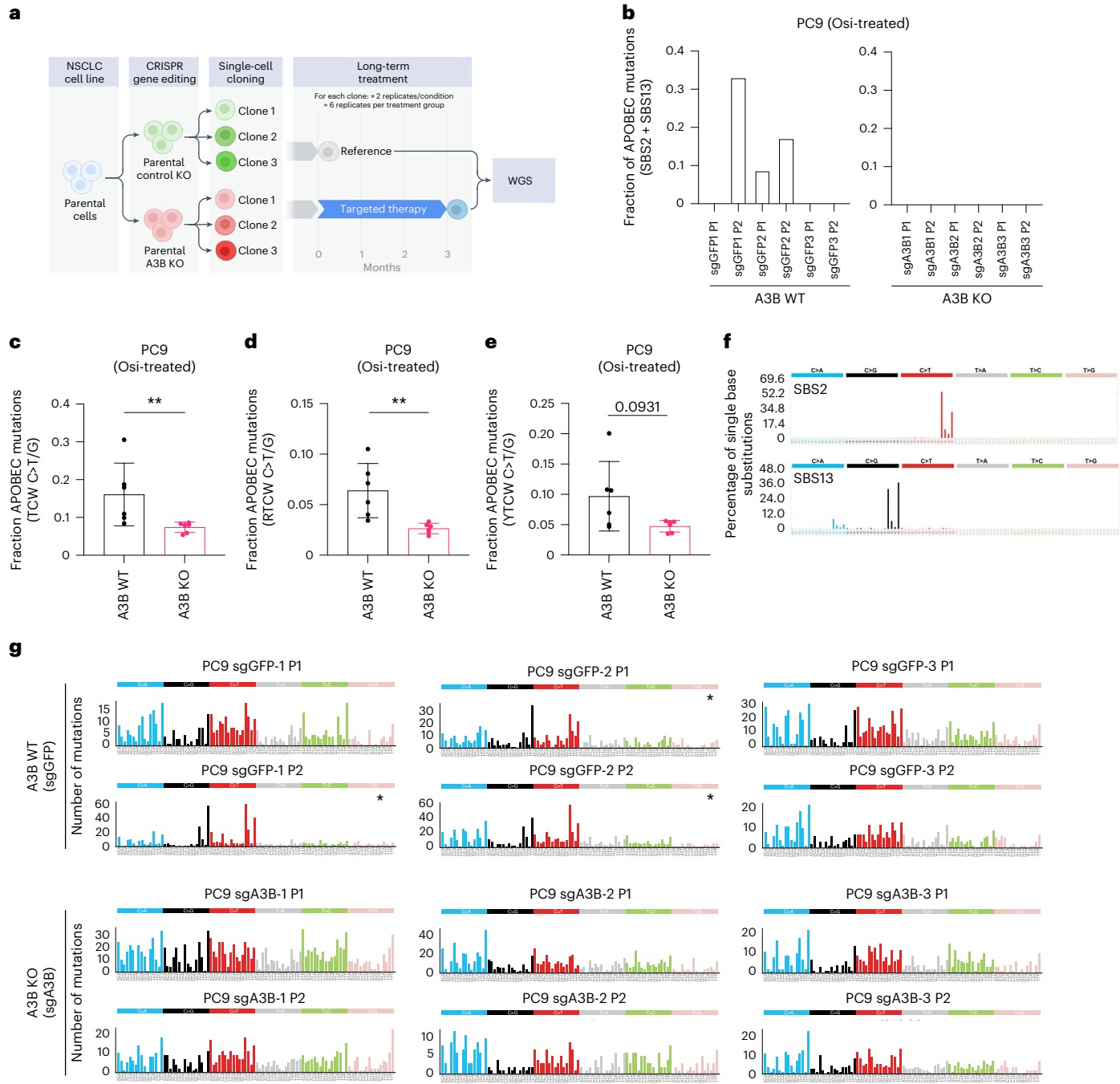

**Fig. 6 | APOBEC3B is required for APOBEC signature accumulation in Osi-treated human NSCLC cell line PC9. a**, Outline of WGS long-term TKI treatment experiment on *APOBEC3B* (*A3B*)-deficient and *A3B*-proficient PC9 single-cell clone lines. Figure created in BioRender.com. **b**, Focused plots showing APOBEC signature (SBS2 + SBS13) burden in the indicated *A3B*-deficient (A3B KO) and *A3B*-proficient (A3B WT) PC9 clones (A3B WT, *n* = 6 biological replicates; A3B KO, *n* = 6 biological replicates). **c**, Fraction of mutations in an APOBEC context (TCW C>T/G) of total mutations per replicate, of Osi-treated A3B WT and A3B KO cells (all data points shown, *n* = 6 biological replicates, mean ± s.d., two-tailed Mann–Whitney test, **P = 0.0043). **d**, Fraction of APOBEC mutations (RTCW C>T/G) of total mutations per replicate Osi-treated A3B WT and A3B KO cells (all data points

shown, *n* = 6 biological replicates, two-tailed Mann–Whitney test, **P = 0.0022). **e**, Fraction of APOBEC mutations (YTCW C>T/G) of total mutations per replicate in Osi-treated A3B WT and A3B KO cells (all data points shown, *n* = 6 biological replicates, two-tailed Mann–Whitney test, *P* = 0.0931). **f**, Profiles of APOBEC-associated signatures SBS2 and SBS13 from the Catalogue of Somatic Mutations in Cancer (COSMIC) (cancer.sanger.ac.uk). **g**, Mutational profiles of A3B KO and A3B WT Osi-treated PC9 cell lines. Mutational profiles are plotted as the number of mutations (*y* axis) at cytosine or thymine bases classified into 96 possible trinucleotide sequence contexts (asterisk indicates cell lines that acquired APOBEC signature during TKI treatment timecourse (SBS2 + SBS13; A3B WT, *n* = 6 biological replicates; A3B KO, *n* = 6 biological replicates)).

(SBS2 + SBS13) acquired were quantified following whole-genome sequencing (WGS; Fig. 6a–g, and Extended Data Fig. 9a). This revealed that only *A3B*-proficient lines gained APOBEC mutation signatures (SBS2 + SBS13) during TKI treatment (Fig. 6b,f,g and Supplementary Table 3). Examination of the fraction of mutations in an APOBEC context

(TCW C>T/G) revealed a significant decrease in *A3B*-deficient lines (Fig. 6c). Examination of APOBEC pentanucleotide sequences[6,32,36,45] in the osimertinib-treated *A3B*-deficient and *A3B*-proficient groups (Fig. 6d,e) revealed significant decreases in the fraction of APOBEC mutations in an A3B-preferred RTCW context in *A3B*-deficient clones, with

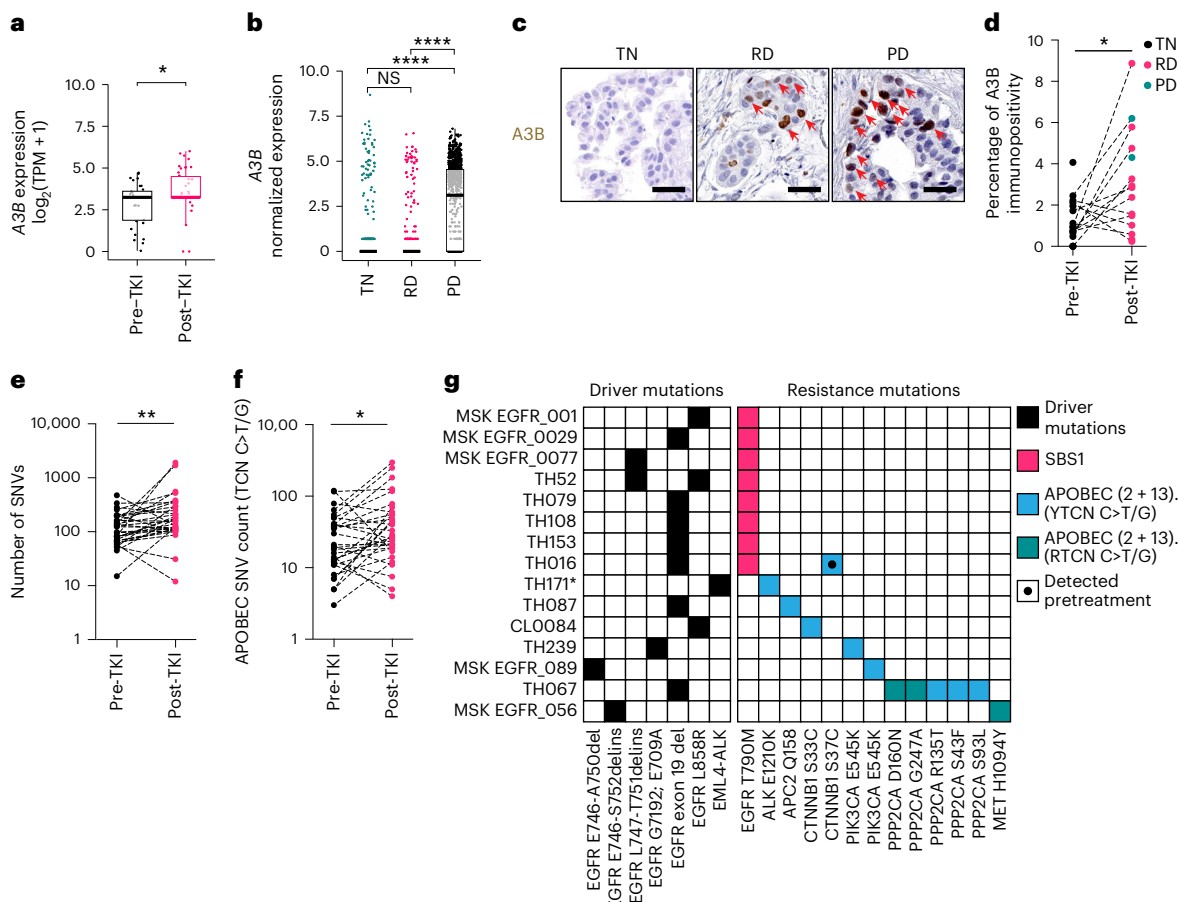

**Fig. 7 | *APOBEC3B* expression and APOBEC-associated mutations are elevated with targeted therapy in patients with NSCLC. a**, *APOBEC3B* (*A3B*) expression levels (batch-corrected transcripts per million (TPM)) measured using RNA-seq analysis in human NSCLC specimens driven by EGFR- and ALK-driver mutations obtained before TKI treatment (pre-TKI, $n = 32$ samples) or post-treatment (post-TKI, $n = 42$ samples; all data points shown, two-sided *t* test, *$P = 0.02$). **b**, APOBEC family member expression measured using single-cell RNA-seq obtained from human NSCLC before TKI treatment (TN), on-treatment at RD or at PD (all data points shown, $n = 762$, 553 and 988 cells per group, respectively, two-sided Wilcoxon test with a Holm correction, ****$P < 2.22 \times 10^{-16}$). **c,d**, Representative images of IHC analysis of A3B protein levels in patients with NSCLC at TN, RD and PD stages. Red arrows indicate positive stained cells (scale bar: 30 μM, **c**)

with IHC quantification of human NSCLC samples pre-TKI ($n = 16$ samples) or post-TKI single agent ($n = 15$ samples; all data points shown, two-sided unpaired *t* test, *$P = 0.0113$, **d**). **e**, Total mutation burden (SNV count) in paired human NSCLC samples pre-TKI or post-TKI ($n = 32$, two-tailed Wilcoxon matched-pairs signed-rank test, **$P = 0.0013$). **f**, APOBEC-associated mutation count in paired human NSCLC samples pre-TKI or post-TKI ($n = 32$, two-tailed Wilcoxon matched-pairs signed-rank test, *$P = 0.0155$). **g**, Mutation signature associated with each putative de novo TKI resistance mutation detected in clinical samples analyzed post-TKI at PD. An asterisk denotes a sample from a patient who has received prior chemotherapy. Boxplots: middle line represents median; lower and upper hinges represent the first and third quartiles; lower and upper whiskers represent smallest and largest values within 1.5× interquartile range from hinges.

---

no significant decrease in mutations in a A3A-preferred YTCW context (Fig. 6d,e). These data suggest that A3B is required for the accumulation of APOBEC mutations during TKI treatment.

To further explore this hypothesis, we analyzed sequencing data for potential TKI resistance mutations in *A3B*-proficient PC9 TKI-resistant clones and found an acquired early stop codon mutation in the tumor suppressor gene *NRXN3* (Q54*)[46,47] in an APOBEC-preferred context (T(C>T)A). The potential impact of this loss-of-function mutation was validated by depleting *NRXN3* (given the early stop codon mutation detected, which is likely a loss-of-function event) in a naïve PC9 lung cancer cell line, which increased levels of phosphorylated AKT, a previously identified convergent feature of EGFR TKI resistance[48], and conferred resistance to EGFR TKI treatment (Extended Data Fig. 9b–d).

**A3B expression and APOBEC-associated mutations are elevated with targeted therapy in NSCLC.** To verify the clinical relevance of our findings, *A3B* expression was examined in several NSCLC clinical datasets (Supplementary Table 2b)[41,49–52]. Bulk RNA-seq of 32 pre-TKI and 42 post-TKI treated (osimertinib/erlotinib/crizotinib/alectinib)

clinical tumor samples revealed a significant increase of *A3B* expression post-TKI relative to pre-TKI samples ($P = 0.011$; Fig. 7a). *A3B* was the only *A3* family member with significantly increased expression post-TKI treatment (Extended Data Fig. 10a). Stratification at TN, RD and PD timepoints revealed a significant expression increase from TN to RD ($P = 0.02$) and an increase approaching significance from TN to PD ($P = 0.057$; Extended Data Fig. 10b). Further validating these observations, single-cell RNA-seq data revealed that *A3B* expression, specifically in tumor cells isolated from clinical specimens, was significantly increased from TN to PD ($P < 0.001$) and from RD to PD ($P < 0.001$; Fig. 7b). Compared to the other *A3* genes, *A3B* expression had the second highest effect scores of all A3 family members as calculated using Cohen's *d* method (TN to PD, $d = 1.048$; RD to PD, $d = 0.953$; Extended Data Fig. 10c). *A3C* expression exhibited the highest effect scores; however, APOBEC activity assays revealed A3C did not contribute to overall activity with TKI treatment (Fig. 4g). Immunohistochemical (IHC) analyses, as performed previously[4], on clinical samples also revealed a significant increase in A3B nuclear protein levels in EGFR TKI-treated tumor samples both at RD and PD timepoints (Fig. 7c,d and Supplementary Table 2c).

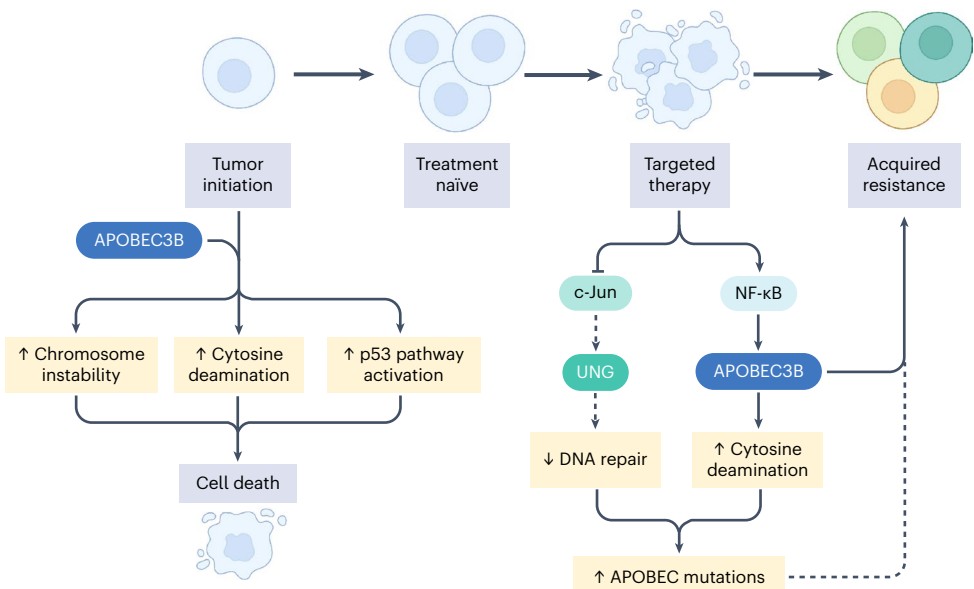

**Fig. 8 | APOBEC3B in EGFR-driven lung tumor evolution.** At tumor initiation, continuous *APOBEC3B* expression and activity induces CIN and p53 pathway activation, resulting in cell death. With targeted therapy, NF-κB induction leads to increased *A3B* expression, fueling TKI resistance. Figure created in BioRender.com.

Demonstrating the clinical effect of TKI treatment on the proportion of mutational signatures, a recently published dataset shows that APOBEC-associated mutation signatures (SBS2 and SBS13) were dominant, defined as the mutational signature with the highest fraction of mutations, in a significantly higher number of osimertinib-resistant samples when compared with naïve samples[53]. To independently test this observation with our own data, WES was performed on paired pre- and post-TKI treated samples obtained from 32 patients (Supplementary Table 4) to quantify mutations acquired following TKI treatment in NSCLC EGFRmut (treated with erlotinib/osimertinib) and ALK fusion (treated with alectinib) clinical samples. This analysis revealed that both the overall mutation burden (SNV count; Fig. 7e) and number of APOBEC-associated mutations (C>T or C>G mutations in a TCN context; Fig. 7f) increased post-treatment.

Next, mutations in an APOBEC-preferred context were identified in genes previously associated with TKI resistance in tumors from patients who had progressed on or shown incomplete response to EGFR inhibitor therapy (Fig. 7g and Supplementary Table 5). These mutations include activating mutations in PIK3CA (E545K)[54], WNT signaling-activating mutations in β-catenin at a glycogen synthase kinase-3β (GSK-3β) phosphorylation site[55], MAPK pathway reactivating-mutations through inactivation of PP2A, a negative regulator of MAPK signaling[56,57], an activating mutation in MET tyrosine kinase domain (H1095Y)[53,58], as well as an ALK inhibitor desensitizing mutation in ALK (E1210K)[59] in the tumors of some patients who had progressed on or shown incomplete response to EGFR or ALK inhibitor therapy. AKT, WNT and MAPK pathway activation have previously been shown to cause EGFR and ALK inhibitor resistance[60–65]. All but one of these APOBEC-associated putative resistance mutations were detected selectively post-treatment, suggesting not only that these mutations are induced by APOBEC (itself engaged) during targeted therapy but also that these variants could promote resistance. All samples containing these APOBEC-associated mutations, except for one, did not harbor a detectable EGFR T790M mutation, which has been reported to be present in ~50–60% of first- and second-generation EGFR TKI-resistant cases[66,67] and arising from a non-APOBEC clock-like mutation signature (SBS1 (ref. 68); Fig. 7g). Altogether, of the resistance mutations in this cohort, 53% (8/15) of mutations were associated with clock-like mutation signature SBS1 and 46% (7/15) of mutations with the APOBEC signatures SBS2 + SBS13, with no other mutational signatures

contributing to putative resistance mutations. In total, 8/32 tumors have APOBEC-associated putative resistance mutations. The observation that APOBEC-mediated mutations in resistance-associated genes detected in post-treatment samples and the EGFR T790M mutation appear to be mutually exclusive suggests that these APOBEC-mediated mutations could be the potential mechanism of resistance to targeted therapy in these patients. These data suggest that APOBEC signatures are a complementary route to acquired TKI therapy resistance, contributing to the diverse mechanisms of resistance that exist[69–71].

Taken together, these data illustrate the diverse effects of A3B at different stages of tumor evolution with or without the selective pressure of therapy. The findings demonstrate multiple roles of A3B, as an inhibitor of tumor progression at initiation, an inducer of APOBEC mutations and a contributor to targeted therapy resistance (Fig. 8).

## Discussion

Our collective findings shed light on the important, context-specific roles of A3B on lung cancer pathogenesis and tumor evolution. Along with other recent findings in the field[5], our data reinforce the concept that targeted therapies can induce adaptive changes that promote resistance[72], including those that are APOBEC-mediated and that may involve multiple APOBEC family members. This A3 induction during therapy might contribute to the development of treatment resistance and appears to be clinically relevant based on our clinical datasets obtained from targeted therapy-treated patients. Additional clinical cohort analyses will be important to conduct as further human tumors obtained from patients on targeted therapy become available.

We demonstrate that the expression of *A3* family members might contribute to resistance in preclinical human and mouse models of lung adenocarcinoma. Although we focus on oncogenic EGFR-driven lung adenocarcinomas, our findings appear to extend to other molecular subsets such as EML4-ALK-driven lung cancer (Fig. 3l and Extended Data Fig. 6b–d) and likely reflect a more general principle of targeted therapy-induced adaptability. While APOBEC has been implicated in drug resistance previously[33,73], our study reveals a distinct mechanism by which targeted cancer therapy is actively responsible for the upregulation of APOBEC via NF-κB-mediated transcriptional induction in response to therapy. Our study further explains the enhanced efficacy of cotreatment with an NF-κB inhibitor compared to EGFR inhibition alone at preventing the emergence of resistance[38].

There are however caveats to our findings (further discussion in Supplementary Note). The mouse models, although helpful for a deeper understanding of the biological effects of enforced A3B expression, are imperfect as *A3B* is expressed from a transgene promoter system. APOBEC3 enzyme expression has also been shown to occur episodically[32], which differs from the constitutive expression of our mouse models. Future studies that reveal the upstream regulators of endogenous mouse APOBEC enzymes could help in the development of better models in future studies.

Our work expands upon prior studies suggesting a potential association between APOBEC-mediated mutagenesis and acquisition of putative resistance mutations in the APOBEC-preferred context during the treatment of EGFR-driven lung cancers[74,75]. Our data suggest that inhibition of APOBEC3 family members could suppress the emergence of one pathway to resistance and thereby improve response to targeted therapy, consistent with the work of others in the field that suggests that multiple APOBEC3 family members including A3B contribute to targeted therapy resistance[5,32], with both A3A and A3B shown to be contributors of mutagenesis[6,32,36,76]. The role of A3B in promoting resistance to TKI is likely multifaceted, and our data do not discount the contribution of other possible parallel cytosine deaminase-independent mechanisms, such as induced CIN[4,77], regulation of cell cycle[22] and regulation of the DNA damage repair pathway[78,79]. Our evidence here and these emerging collective findings[5,33,80,81] suggest that endogenous drivers of mutagenesis have diverse roles that are both detrimental and beneficial to tumor evolution depending on the context of tumor pathogenesis and treatment.

## Online content

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

Deborah R. Caswell [1,46] ✉, Philippe Gui[2,46], Manasi K. Mayekar[2,46], Emily K. Law [3], Oriol Pich [1], Chris Bailey[1], Jesse Boumelha [4], D. Lucas Kerr [2], Collin M. Blakely [2], Tadashi Manabe [2], Carlos Martinez-Ruiz[5,6], Bjorn Bakker [1], Juan De Dios Palomino Villcas[7], Natalie I. Vokes [8,9], Michelle Dietzen [1,5,6], Mihaela Angelova [1], Beatrice Gini [2], Whitney Tamaki [2], Paul Allegakoen[2], Wei Wu [2], Timothy J. Humpton [10,11,12], William Hill[1], Mona Tomaschko[4], Wei-Ting Lu [1], Franziska Haderk[2], Maise Al Bakir [1], Ai Nagano [1], Francisco Gimeno-Valiente[6], Sophie de Carné Trécesson[4], Roberto Vendramin[1], Vittorio Barbè[1], Miriam Mugabo[6], Clare E. Weeden[1], Andrew Rowan[1], Caroline E. McCoach[13], Bruna Almeida[14,15], Mary Green[16], Carlos Gomez[2], Shigeki Nanjo [2], Dora Barbosa[2], Chris Moore[4], Joanna Przewrocka[1], James R. M. Black [1,5,6], Eva Grönroos [1], Alejandro Suarez-Bonnet [16,17], Simon L. Priestnall [16,17], Caroline Zverev[18], Scott Lighterness[18], James Cormack[18], Victor Olivas[2], Lauren Cech [2], Trisha Andrews[2], Brandon Rule [19], Yuwei Jiao[20], Xinzhu Zhang[20], Paul Ashford [21], Cameron Durfee [22], Subramanian Venkatesan[1], Nuri Alpay Temiz [23,24], Lisa Tan[2], Lindsay K. Larson[3], Prokopios P. Argyris[3,25,26], William L. Brown [3], Elizabeth A. Yu[2,27], Julia K. Rotow[28], Udayan Guha [29,30], Nitin Roper[31], Johnny Yu[32], Rachel I. Vogel [33], Nicholas J. Thomas [2], Antonio Marra[34], Pier Selenica[35], Helena Yu [36,37], Samuel F. Bakhoum[38,39], Su Kit Chew[1], Jorge S. Reis-Filho [36], Mariam Jamal-Hanjani [6,40,41], Karen H. Vousden [10], Nicholas McGranahan[5,6], Eliezer M. Van Allen [42], Nnennaya Kanu[6], Reuben S. Harris [22,43], Julian Downward [4], Trever G. Bivona [44,45] ✉ & Charles Swanton [1,6]

[1]Cancer Evolution and Genome Instability Laboratory, The Francis Crick Institute, London, UK. [2]Department of Medicine, University of California, San Francisco, San Francisco, CA, USA. [3]Department of Biochemistry, Molecular Biology and Biophysics, University of Minnesota, Minneapolis, MN, USA. [4]Oncogene Biology Laboratory, The Francis Crick Institute, London, UK. [5]Cancer Genome Evolution Research Group, University College London, Cancer Institute, London, UK. [6]Cancer Research UK Lung Cancer Centre of Excellence, UCL Cancer Institute, London, UK. [7]Core Research Laboratory, ISPRO, Florence, Italy. [8]Department of Thoracic and Head and Neck Medical Oncology, The University of Texas MD Anderson Cancer Center, Houston, TX, USA. [9]Department of Genomic Medicine, The University of Texas MD Anderson Cancer Center, Houston, TX, USA. [10]p53 and Metabolism Laboratory, The Francis Crick Institute, London, UK. [11]CRUK Beatson Institute, Glasgow, UK. [12]Glasgow Caledonian University, Glasgow, UK. [13]Genentech Inc, South San Francisco, CA, USA. [14]The Roger Williams Institute of Hepatology, Foundation for Liver Research, London, UK. [15]Faculty of Life Sciences & Medicine, King's College London, London, UK. [16]Experimental Histopathology, The Francis Crick Institute, London, UK. [17]Department of Pathobiology & Population Sciences, The Royal Veterinary College, London, UK. [18]Biological Research Facility, The Francis Crick Institute, London, UK. [19]Cursorless, London, UK. [20]Novogene Europe, Cambridge, UK. [21]Institute of Structural and Molecular Biology, University College London, London, UK. [22]Department of Biochemistry and Structural Biology, University of Texas Health San Antonio, San Antonio, TX, USA. [23]Institute for Health Informatics, University of Minnesota, Minneapolis, MN, USA. [24]Masonic Cancer Center, University of Minnesota, Minneapolis, MN, USA. [25]School of Dentistry, University of Minnesota, Minneapolis, MN, USA. [26]College of Dentistry, Ohio State University, Columbus, OH, USA. [27]Sutter Health Palo Alto Medical Foundation, Department of Pulmonary and Critical Care, Mountain View, CA, USA. [28]Lowe Center for Thoracic Oncology, Dana-Farber Cancer Institute, Boston, MA, USA. [29]Thoracic and GI Malignancies Branch, NCI, NIH, Bethesda, MD, USA. [30]NextCure Inc., Beltsville, MD, USA. [31]Developmental Therapeutics Branch, Center for Cancer Research, National Cancer Institute, National Institutes of Health, Bethesda, MD, USA. [32]Biomedical Sciences Program, University of California, San Francisco, San Francisco, CA, USA. [33]Department of Obstetrics, Gynecology and Women's Health, University of Minnesota, Minneapolis, MN, USA. [34]Division of Early Drug Development for Innovative Therapy, European Institute of Oncology IRCCS, Milan, Italy. [35]Department of Pathology, Memorial Sloan Kettering Cancer Center, New York City, NY, USA. [36]Memorial Sloan Kettering Cancer Center, New York City, NY, USA. [37]Department of Medicine, Weill Cornell College of Medicine, New York City, NY, USA. [38]Human Oncology and Pathogenesis Program, Memorial Sloan Kettering Cancer Center, New York City, NY, USA. [39]Department of Radiation Oncology, Memorial Sloan Kettering Cancer Center, New York City, NY, USA. [40]Cancer Metastasis Laboratory, University College London Cancer Institute, London, UK. [41]Department of Medical Oncology, University College London Hospitals, London, UK. [42]Department of Medical Oncology, Dana-Farber Cancer Institute, Boston, MA, USA. [43]Howard Hughes Medical Institute, University of Texas Health San Antonio, San Antonio, TX, USA. [44]Departments of Medicine and Cellular and Molecular Pharmacology, Helen Diller Family Comprehensive Cancer Center, University of California, San Francisco, San Francisco, CA, USA. [45]Chan Zuckerberg Biohub, San Francisco, CA, USA. [46]These authors contributed equally: Deborah R. Caswell, Philippe Gui, Manasi K. Mayekar. ✉e-mail: deborah.caswell@crick.ac.uk; trever.bivona@ucsf.edu

## Methods

### Cell line and growth assays

Cell lines were grown in Roswell Park Memorial Institute-1640 medium (RPMI-1640) with 1% penicillin–streptomycin (10,000 U ml⁻¹) and 10% FBS or in Iscove's modified Dulbecco's medium (IMDM) with 1% penicillin–streptomycin (10,000 U ml⁻¹), L-glutamine (200 mM) and 10% FBS in a humidified incubator with 5% $CO_2$ maintained at 37 °C. Drugs used for treatment except PBS-1086 (ref. 38) were purchased from Selleck Chemicals or MedKoo Biosciences. For growth assays, cells were exposed to DMSO or the indicated drugs for indicated durations in six-well or 96-well plates and assayed using crystal violet staining or Celltiter-Glo luminescent viability assay (Promega) according to the manufacturer's instructions.

### Deriving clonal populations and generating *APOBEC3B* KO cells

Clonal cells were derived by sorting single cells into 96-well plates and expanding them over a few weeks. We then derived pools of one of the clones expressing either a green fluorescent protein (GFP)-targeting or *A3B*-targeting guide along with CRISPR/Cas9 by lentiviral transduction as done in a previously published study[82]. *A3B* gRNA target sequences, designed by the Zhang Lab[83], were subcloned into the lentiCRISPR v2 plasmid (Addgene, 52961; a gift from F. Zhang)[83] and the one that showed better *A3B* depletion was selected for further analysis.

### Transductions and transfections

Hek293T cells were cotransfected with lentiviral packaging plasmids pCMVdr8 and pMD2.G plasmid, along with the plasmid of interest using FuGENE 6 Transfection Reagent (Promega). *APOBEC3B* shRNA was purchased from Sigma (TRCN0000142875). Cells were transduced with 1:1 diluted lentivirus for 1–2 d and selected with antibiotic marker (puromycin). siRNAs were purchased from GE Healthcare Dharmacon and transfected using Lipofectamine RNAi Max according to the manufacturer's protocol, and the cells were collected within 48 h of transfection for subsequent assays.

### RT−qPCR

Total RNA was extracted using GeneJet RNA purification kit (Thermo Fisher Scientific) or RNeasy Mini kit (Qiagen), and cDNA was synthesized from it using sensiFast cDNA Synthesis Kit or High-Capacity cDNA Reverse Transcription Kit (Applied Biosystems) in accordance with the manufacturer's instructions. qPCR reactions were performed using PowerUP SYBR Green Master Mix (Applied Biosystems) or TaqMan Universal PCR Master Mix (Applied Biosystems) and previously validated primers[84] (PrimerBank) on a QuantStudio. Glyceraldehyde-3-phosphatase dehydrogenase (GAPDH), actin, 18S RNA or β2-microglobulin were used as reference genes. The following primers were used for p53 pathway activation: actin: Mm02619580_g1, Bax: Mm00432051_m1, Cdkn1a/p21: Mm04205640_g1, Mdm2: Mm01233138_m1, Pmaip1/Noxa: Mm00451763_m1 and Sesn2: Mm00460679_m1. Data were analyzed using QuantStudio 12K Flex Software (v1.3) and GraphPad Prism.

### Western blot assay

Whole-cell extracts were collected in RIPA buffer containing protease and phosphatase inhibitors followed by sonication and centrifugation for clarification of extracts. Nuclear-cytoplasmic extracts were collected as described previously with 0.1% nonidet P-40 (NP-40) in PBS[85]. Extracts were quantified using Lowry assay, run on 4–15% Criterion TGX Gels (Bio-Rad) and transferred to a nitrocellulose membrane with Trans-Blot Turbo RTA Midi Nitrocellulose Transfer Kit (Bio-Rad). Membranes were blocked in 3% milk in tris-buffered saline with 0.1% Tween 20 (TBST), incubated with primary antibody overnight followed by secondary antibody, either horse radish peroxidase (HRP)-conjugated or fluorescently labeled, for 1–2 h and imaged on a LI-COR imager or

ImageQuant LAS 4000 (GE HealthCare). Anti-APOBEC3B (5210-87-13)[86] and anti-UNG[28] antibodies were kindly provided by R. Harris, and anti-GAPDH antibody (sc-59540) was purchased from Santa Cruz Biotechnology. Anti-EGFR (4267), anti-phospho-EGFR (Y1068, 3777 or 2236), anti-STAT3 (9139), anti-phospho-STAT3 (Y705, 9145), anti-AKT (2920), anti-phospho-AKT (S473, 4060), anti-phospho-ERK (T202, Y204; 4370 or 9106), anti-ERK (9102), anti-RELA (8242), anti-RELB (4922), anti-HSP90 (4874), anti-TUBB (2146) and anti-histone H3 (9715) were purchased from Cell Signaling Technology (CST). All primary antibodies were used at a dilution of 1:1,000.

### Enzymatic assays

APOBEC assays were performed by incubating nuclear extracts from rapid efficient and practical (REAP) method[58] or whole-cell extracts with the following DNA oligo substrates (Integrated DNA Technologies, IDT): 5′-ATT ATT ATT AT**T CA**A ATG GAT TTA TTT ATT TAT TTA TTT ATT T-FAM-3′ using established protocols[28,35]. Upon completion of the reactions, they were heated at 95 °C for 5 min after the addition of TBE-urea buffer (Novex) and immediately run on a 15% TBE-urea gel (Bio-Rad) and imaged using Cy2 filter on ImageQuant LAS 4000.

### Subcutaneous tumor xenografts and PDX studies

All animal experiments were conducted under University of California, San Francisco (UCSF) Institutional Animal Care & Use Committee (IACUC)-approved animal protocols. PC9 and H1975 tumor xenografts were generated by injection of 1 million cells in a 1:1 mixture of matrigel and PBS into 6- to 8-week-old female non-obese diabetic/severe combined immunodeficiency disease (NOD/SCID) mice. Once the tumors grew to ~100 mm³, the mice were treated with vehicle or 5 mg kg⁻¹ osimertinib once daily by oral gavage and the tumors were collected on day 4 for western blot analysis. PDX was generated as indicated in a previous study[38]. Tumors were passaged in SCID mice, treated with 25 mg kg⁻¹ erlotinib once daily by oral gavage once they reached ~400 mm³ and collected on day 2.

### Mouse strains and tumor induction and treatment

The Cre-inducible *Rosa26::LSL-APOBEC3Bi* mice and *Rosa26::CAG-LSL-APOBEC3Bi-E255A* are described in refs. 20,23. The *TetO-EGFR^{L858R}; Rosa26^{LNL-tTA}* (E) and *CCSP-rtTA;TetO-EGFR^{L858R};Rosa26^{CreER(T2)}* mice have been described in refs. 11,12,87,88. All mice were purified C57BL/6J mice, aged between 8 and 20 weeks, with a mixed sex ratio for each experiment (Supplementary Table 6). Tumors were initiated in *E*, *EA3B*, *EP* and *EPA3B* mice by intratracheal infection with adenoviral vectors expressing Cre recombinase as described[89]. Adenoviral-Cre (Ad-Cre-GFP) was from the University of Iowa Gene Transfer Core. Tumors were initiated in *EA3Bi* mice using chow containing doxycycline (625 ppm) obtained from Harlan-Teklad. All animal-regulated procedures were approved by the Francis Crick Institute BRF Strategic Oversight Committee that incorporates the Animal Welfare and Ethical Review Body and conformed with the UK Home Office guidelines and regulations under the Animals (Scientific Procedures) Act 1986 including Amendment Regulations 2012. To assess the recombination efficiency of the LSL allele upstream of APOBEC3B, PCR primers targeting the R26 site, the LSL cassette and the APOBEC3B transgene were used as described[20]. Erlotinib was purchased from Selleckchem (erlotinib, Osi-744), dissolved in 0.3% methylcellulose and administered intraperitoneally at 25 mg kg⁻¹, 5 d a week. Tamoxifen was administered by oral gavage three times in 1 week at 2–4 d intervals (three injections total). Mice received tamoxifen at 150 mg kg⁻¹ dissolved in sunflower oil.

### Assessment of recombination efficiency

PCR was performed to assess the recombination of the LSL cassette upstream of the *A3B* allele in six tumors collected at progression. Five of six (5/6) of the tumors had a recombination efficiency above 90%, and one tumor of six was unrecombined. This rate of recombination aligns

with the rate of recombination observed by IHC staining at 3 months and at termination and suggests that a lack of recombination of the LSL cassette upstream of the A3B transgene explains A3B-negative tumors.

## Micro-computed tomography (micro-CT) imaging

Mice were anesthetized with isoflurane/oxygen for no more than an hour each and minimally restrained during imaging (~8 to 10 min). Mice were then observed and, if necessary, placed in cages in a recovery chamber/rack until they regained consciousness and started to feed. Tumor burden was quantified by calculating the volume of visible tumors using AnalyzeDirect.

## Histological preparation and IHC staining

Tissues were fixed in 10% formalin overnight and transferred to 70% ethanol until paraffin embedding. IHC was performed using the following primary antibodies: EGFR[L858R] mutant specific (CST, 3197 and 43B2), APOBEC3B (5210-87-13)[86], Ki67 (Abcam, Ab15580), Caspase 3 (R&D (Bio-Techne), AF835), p-Histone H2AX (Sigma-Aldrich, 05-636), Phospho-Histone H3 (Ser10; CST, 9706), CD4 (Abcam, ab183685; EPR19514), CD8 (Thermo Fisher Scientific, 14-0808-82; 4SM15) and UNG (Novus Biologicals, NB600-1031). Sections were developed with 3,3′-Diaminobenzidine (DAB) and counterstained with hematoxylin. Staining for p53 (Leica, NCL-L-p53-CM5p) was performed on a Dako Autostainer Link 48 (Agilent) as previously described[90]. The number of EGFR[L858R], APOBEC3B, Ki67, Caspase 3 and gH2AX-positive cells were quantified using QuPath.

## Evaluation of chromosome missegregation errors in hematoxylin and eosin (H&E)- and/or phospho-histone H3-stained samples

Lung sections were evaluated for anaphases with chromosome missegregation events using a ×100 objective light microscope. For *E* and *EA3B* mice at early and late timepoints, the percentage of missegregation errors was calculated and averaged across all mice using the harmonic mean. For *EA3B* mice, the percent error was normalized to an A3B recombination efficiency of 82% based on observed recombination efficiency observed (Extended Fig. 4). For *E* and *EA3Bi* mice with subclonal A3B expression, normalization for the recombination efficiency was not possible, so the percentage of missegregation errors was calculated based on the number of errors versus normal anaphases observed.

## Mouse tumor processing

Frozen tumor tissue was cut into pieces and lysed in RLT Buffer with β-mercaptoethanol. TissueRuptor was used for disruption and homogenization of tissue. Lysate was added to a QIAshredder tube and centrifuged at full speed for 1 min. The homogenized solution was then added to AllPrep DNA spin columns (Qiagen AllPrep DNA/RNA Mini Kit, 80204).

## Histopathological examination of mouse

Four micrometers thick, formalin-fixed, paraffin-embedded (FFPE) sections from lung lobes were stained with H&E and examined by two board-certified Veterinary Pathologists (A.S.B. and S.L.P.). Histopathological assessment was performed blind to experimental grouping using a light microscope (Olympus, BX43). Tissue sections were examined individually, and in case of discordance in diagnosis, a consensus was reached using a double-head microscope.

Proliferative lesions were diagnosed as alveolar hyperplasia, bronchioloalveolar adenoma and well-differentiated, moderately or poorly differentiated bronchioloalveolar adenocarcinoma. Sections were histopathologically assessed and graded for the presence and type of proliferative epithelial lung lesions using the International Harmonization of Nomenclature and Diagnostic Criteria for Lesions (INHAND) guide for nonproliferative and proliferative lesions of the respiratory tract of the mouse[91].

## WES−mouse data

WES was performed by the Advanced Sequencing Facility at the Francis Crick Institute using the Human Core Exome Kit (Twist BioScience) for library preparation and SureSelectXT Mouse All Exon, 16, Kit (Agilent) for library preparation, respectively. Sequencing was performed on HiSeq 4000 platforms.

## RNA-seq−mouse data

RNA-seq was performed by the Advanced Sequencing Facility at the Francis Crick Institute using the KAPA mRNA HyperPrep Kit (KK8581−96 Libraries) and KAPA Dual-Indexed Adapters (Roche, KK8720). Sequencing was performed on HiSeq 4000 platforms. The processed FASTQ files were mapped to mm10 reference genome using the STAR (version 2.4) algorithm, and transcript expressions were quantified using the RSEM (version 1.2.29) algorithm with the default parameters. The read counts were used for downstream analysis.

## Alignment−mouse

All samples were demultiplexed, and the resultant FASTQ files aligned to the mm10 mouse genome, using BWA-MEM (BWA, v0.7.15). Deduplication was performed using Picard (v2.1.1; http://broadinstitute.github.io/picard). Quality control metrics were collated using FASTQC (v0.10.1; http://www.bioinformatics.babraham.ac.uk/projects/fastqc/), Picard and GATK (v3.6). SAMtools (v1.3.1) was used to generate mpileup files from the resultant BAM files. Thresholds for base phred score and mapping quality were set at 20. A threshold of 50 was set for the coefficient of downgrading mapping quality, with the argument for base alignment quality calculation being deactivated. The median depth of coverage for all samples was 92× (range: 58–169×).

## Variant detection and annotation−mouse

Variant calling was performed using VarScan2 (v2.4.1), MuTect (v1.1.7) and Scalpel (v0.5.4)[92–94].

The following argument settings were used for variant detection using VarScan2:

--min-coverage 8 --min-coverage-normal 10 --min-coverage-tumor 6 --min-var-freq 0.01 --min-freq-for-hom 0.75 --normal-purity 1 --p-value 0.99 --somatic-p-value 0.05 --tumor-purity 0.5 --strand-filter 0

For MuTect, only 'PASS' variants were used for further analyses. Except for allowing variants to be detected down to a variant allele frequency (VAF) of 0.001, default settings were used for Scalpel insertion/deletion detection.

To minimize false positives, additional filtering was performed. For single-nucleotide variants (SNVs) or dinucleotides detected by VarScan2, a minimum tumor sequencing depth of 30, VAF of 5%, variant read count of 5 and a somatic *P* value < 0.01 were required to pass a variant. For variants detected by VarScan2 between 2% and 5% VAF, the mutation also needs to be detected by MuTect.

As for insertions/deletions (INDELs), variants need to be passed by both Scalpel (PASS) and VarScan2 (somatic *P* < 0.001). A minimum depth of 50×, 10 alt reads and VAF of 2% were required.

For all SNVs, INDELs and dinucleotides, any variant also detected in the paired germline sample with more than five alternative reads or a VAF greater than 1% was filtered out.

The detected variants were annotated using Annovar[95].

## Functional annotation of SNVs−mouse

Mouse gene mutation callings from WES were parsed with some modifications including genomic coordinates (removing 'chr' before chromosomal numbers, only 'SNV' was selected). The modified files were fed into Protein Variation Effect Analyzer (PROVEAN)[96–98] software tool (http://provean.jcvi.org/index.php) to predict whether an amino acid substitution has an impact on the biological function of a protein (Sorting Intolerant From Tolerant, SIFT score). The predict files were merged with original files at gene level annotation using the R program.

## Human *EGFR* transgene amplicon sequencing of mouse

FASTQ files were aligned to hg19 obtained from the GATK bundle (v2.8) using BWA-MEM (BWA, v0.7.15)[99,100]. Analyses were performed using R (v3.3.1) and deepSNV (v1.18.1)[101]. The median depth of coverage of sequenced EGFR exons (19,20,21) was 5290× (range: 2,238–8,040). Variants associated with resistance to EGFR TKIs were queried using deep-SNV's bam2R function, with the arguments $q = 20$ and $s = 2$. The variants explored include the following: T790M, D761Y, L861Q, G796X, G797X, L792X and L747S. L858R was identified in every sequenced sample.

## Generation of EGFR^L858R mutant mouse tumor cell lines

A portion of mouse lung tumor was dissected (1/3 to 1/2 of the original tumor depending on size) and cut into small pieces with scissors. Pieces were then digested for 30 min at 37 °C while rotating at full speed in digestion media (1,400 µl HBSS-free w/o Ca^2+, 200 µl Collagenase IV and 40 U ml^−1 DNase). Tumor cells were pelleted down in a centrifuge (1,100 r.p.m. for 4 min) and resuspended in IMDM supplemented with 1% penicillin–streptomycin solution (10,000 U ml^−1), L-glutamine (200 mM) and 10% FBS. This cell suspension was then plated in a 10-cm plate and passaged over a period of 1–3 months until consistent growth was observed.

## Generation of TKI-resistant mouse or human tumor cell lines

TKI naïve cell lines were cultured in increasing levels of erlotinib or osimertinib using a dose-escalation protocol from 100 nM to 1 µM when cells were growing with minimal cell death.

## Mutational and SCNA ITH calculations for TRACERx data

SCNA ITH was calculated by dividing the percentage of the genome harboring heterogeneous SCNA events, that is, those events that were not present in every region, by the percentage of the genome involved in any SCNA event in each tumor[25].

## Cell line whole-genome mutational signature analysis

Sequences were aligned to the human genome (hg38) using the Burrows-Wheeler Aligner (version 0.7.17). PCR duplicates were removed using Picard (version 2.18.16). Reads were locally realigned around indels using GATK3 (version 3.6.0) tools RealignerTargetCreator to create intervals, followed by IndelRealigner on the aligned BAM files. MuTect2 from GATK3 (version 3.6.0) was used in tumor/normal mode to call mutations in test versus control cell lines. SNVs that passed the internal GATK3 filter with read depths over 30 reads at called positions, at least 4 reads in the alternate mutation call and an allele frequency greater than 0.05 were used for downstream analysis. Mutational profile plots in Fig. 6g were plotted using the deconstructSigs R package[102].

## DNA and RNA isolation from cell line models for sequencing

DNA or RNA were extracted from frozen cell pellets using Qiagen's DNeasy Blood and Tissue Kit or Qiagen's RNeasy MINI Kit, respectively, as per the manufacturer's instructions. The isolated DNA or RNA was quantified and qualitatively assessed using a Qubit Fluorometer (Thermo Fisher Scientific) and a Bioanalyzer (Agilent), as per the manufacturer's instructions. The DNA or RNA were then sent to BGI for WGS (30×) or Novogene for mRNA or WES.

## Cell cycle analysis

Cell cycle analysis was performed by propidium iodide (PI) staining. Briefly, PC9 cells were treated for 24 h with DMSO, 2 µM osimertinib or 1 µM palbociclib and then fixed in ice-cold 70% ethanol and stained with a 50 µg ml^−1 PI (MilliporeSigma, P4864) + 0.1% Triton X-100 (MilliporeSigma, X100) solution. PI fluorescence was then measured on a flow cytometer (BD FACSAria II).

## Human participants

All patients gave informed written consent for the collection of clinical correlates, tissue collection and research testing under institutional review board (IRB)-approved protocols (CC13-6512 and CC17-658, NCT03433469). Patient demographics are listed in Supplementary Tables 2a–c, 4, and 5a,b. Patient studies were conducted according to the Declaration of Helsinki, the Belmont Report and the U.S. Common Rule.

## Studies with specimens from patients with lung cancer

Frozen or FFPE tissues from patients with lung cancer for DNA or RNA sequencing (bulk and single cell) studies were processed and sequenced as described previously[41,60]. Classification of response was based on RECIST criteria. Some of these biopsies were subjected to WES at the QB3-Berkley Genomics for which library preparation was performed using IDT's xGen exome panel. For additional specimens, tumor DNA from FFPE tissues and matched nontumor from blood aliquots or stored buffy coats were collected as part of the UCSF biospecimen resource program (BIOS) in accordance with UCSF's IRB-approved protocol. DNA from blood aliquots was isolated at the BIOS. Other nontumor samples and FFPE tumor tissues were sent for extraction and assessment of quality and quantity to Novogene, and those meeting the required sample standards were subjected to WES at Novogene's sequencing facility.

## Mutation analysis

Paired-end reads were aligned to the hg19 human genome using the Picard pipeline (https://gatk.broadinstitute.org/). A modified version of the Broad Institute Getz Lab CGA WES Characterization pipeline (https://docs-google-com.ezp-prod1.hul.harvard.edu/document/d/1VO2kX_fgfUd0x3mBS9NjLUWGZu794WbTepBel3cBg08) was used to call, filter and annotate somatic mutations. Specifically, SNVs and other substitutions were called with MuTect (v1.1.6)[93]. Mutations were annotated using Oncotator[103]. MuTect mutation calls were filtered for 8-OxoG artifacts, and artifacts were introduced through the formalin fixation process (FFPE) of tumor tissues[66]. Indels were called with Strelka (v1.0.11). MuTect calls and Strelka calls were further filtered through a panel of normal samples (PoN) to remove artifacts generated by rare error modes and miscalled germline alterations[93]. To pass quality control, samples were required to have <5% cross-sample contamination as assessed with ContEst[93]; mean target coverage of at least 25× in the tumor sample and 20× in the corresponding normal as assessed using GATK3.7 DepthOfCoverage and a percentage of tumor-in-normal of <30% as determined by deTiN[104]. This pipeline was modified for analysis of cell lines rather than tumor-normal pairs as follows: indels were called through MuTect2 alone rather than Strelka; deTiN was not performed and a common variant filter was applied to exclude variants present in the Exome Aggregation Consortium if at least ten alleles containing the variant were present across any subpopulation, unless they appeared in a list of known somatic sites[105,106].

## Mutational signature analysis

Active mutational processes[107] were determined using the deconstructSigs R package[63], with a signature contribution cutoff of 6%. This cutoff was chosen because it was the minimum contribution value required to obtain a false-positive rate of 0.1% and a false-negative rate of 1.4% per the authors' in silico analysis and is the recommended cutoff[102]. Samples with <10 mutations were excluded from analysis due to poor signature discrimination with only a few mutations, and a sample with less than 15 d of exposure to TKI therapy was excluded because it is too short a time to accumulate detectable mutations due to therapy. For TRACERx data analysis, data processing was performed in the R statistical environment version ≥3.3.1.

## RNA-seq analyses

PDX tissue and mouse tumor cell line RNA extractions were carried out using an RNeasy Micro Kit (Qiagen). RNA-seq was performed on PDX tissue using replicate samples on the Illumina HiSeq 4000, paired-end

100-bp reads at the Center for Advanced Technology (UCSF). For the differential gene expression analysis, DESeq program was used to compare controls to erlotinib samples as previously described[108].

RNA-seq samples from patients and cell lines were sequenced by Novogene (https://en.novogene.com/) with paired-end sequencing (150 bp in length). There were ~20 million reads for each sample. The processed FASTQ files were mapped to the hg19 reference genome using the STAR (version 2.4) algorithm, and transcript expressions were quantified using the RSEM (version 1.2.29) algorithm. The default parameters in the algorithms were used. The normalized transcript reads (TPM) were used for downstream analysis. Gene set enrichment analysis was performed using GSEA software[109].

For single-cell RNA-seq analyses, the data from a previously published study (all cancer cells from patients with advanced lung cancer) were used and analyzed in a similar manner[41]. All cells used are identified as malignant by marker expression and CNV inference and originated in from various biopsy sites (adrenal, liver, lymph node, lung and pleura/pleural fluid). Nonparametric, pairwise comparisons (Wilcoxon rank-sum test) were used to determine the statistical significance of the pairwise comparisons of different timepoints for their average scaled expression.

### Statistical analysis

One-way or two-way ANOVA test with Holm–Sidak correction for multiple comparisons (>2 groups) or two-tailed $t$ test (2 groups) were used to determine the statistical significance of the differences between groups for RT–qPCR, growth and enzymatic assays and bulk RNA-seq analysis. Normality of IHC and micro-CT data was determined using multiple testing methods (Anderson–Darling test, D'Agostino–Pearson test, Shapiro–Wilk test and Kolmogorov–Smirnov test). A two-sided $t$ test or two-sided Mann–Whitney test was used for IHC and micro-CT data depending on the normality tests to determine the statistical significance of the differences between groups. Analysis for these assays was done using GraphPad Prism.

### Reporting summary

Further information on research design is available in the Nature Portfolio Reporting Summary linked to this article.

### Data availability

The WES data and RNA-seq data (from the TRACERx study) used during this study have been deposited at the European Genome-phenome Archive (EGA), which is hosted by the European Bioinformatics Institute and the Center for Genomic Regulation under the accession codes EGAS00001006494 and EGAS00001006517, respectively, is under controlled access due to its nature and commercial licenses. Specifically, data are available through the Cancer Research UK and University College London Cancer Trials Center (ctc.tracerx@ucl.ac.uk) for academic noncommercial research purposes only and are subject to review of a project proposal by the TRACERx data access committee, entering into an appropriate data access agreement and subject to any applicable ethical approvals. A response to the request for access is typically provided within ten working days after the committee has received the relevant project proposal and all other required information. The WES data of tumor-derived cell lines shown in Extended Data Fig. 3 are available at the European Nucleotide Archive (ENA) with the identifier PRJEB67640 (ERP152649). The WGS data of PC9 cell lines shown in Fig. 6 are available at the ENA with the identifier PRJEB67559 (ERP152586). For the single-cell RNA-seq analyses shown in Extended Data Fig. 10b,c, the data from a previously published study (all advanced lung cancer cell data) were used and analyzed in a similar manner[41]. These data are available in the National Center for Biotechnology Information (NCBI) BioProject ID PRJNA591860. The RNA-seq data for Extended Data Fig. 10a were from a previously published study[38]. These data are available at NCBI GEO under accession GSE65420. Clinical sample RNA-seq and WES sequencing data are available in NCBI Bio-Project ID PRJNA1029563. Source data are provided with this paper.

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

## Acknowledgements

C.S. is a Royal Society Napier Research Professor (RSRP\R\210001). His work is supported by the Francis Crick Institute which receives its core funding from Cancer Research UK (CRUK) (CC2041), the UK Medical Research Council (CC2041) and the Wellcome Trust (CC2041). For Open Access, the author has applied a CC BY public copyright license to any Author Accepted Manuscript version arising from this submission. C.S. is funded by Cancer Research UK (TRACERx (C11496/A17786), PEACE (C416/A21999) and CRUK Cancer Immunotherapy Catalyst Network); Cancer Research UK Lung Cancer Center of Excellence (C11496/A30025); the Rosetrees Trust, Butterfield and Stoneygate Trusts; Novo Nordisk Foundation (ID16584); Royal Society Professorship Enhancement Award (RP/EA/180007); National Institute for Health Research (NIHR) University College London Hospitals Biomedical Research Center; the Cancer Research UK-University College London Center; Experimental Cancer Medicine Center; the Breast Cancer Research Foundation (United States, BCRF-22-157); Cancer Research UK Early Detection and Diagnosis Primer Award (grant EDDPMA-Nov21/100034) and the Mark Foundation for Cancer Research Aspire Award (grant 21-029-ASP). This work was supported by a Stand Up To Cancer-LUNGevity-American Lung Association Lung Cancer Interception Dream Team Translational Research Grant (grants SU2C-AACR-DT23-17 to S.M. Dubinett and A.E. Spira). Stand Up To Cancer is a division of the Entertainment Industry Foundation. Research grants are administered by the American Association for Cancer Research, the Scientific Partner of SU2C. C.S. is in receipt of an ERC Advanced Grant (PROTEUS) from the European Research Council under the European Union's Horizon 2020 research and innovation program (grant 835297). This project is supported by the NIH/NCI U54CA224081, R01CA169338, R01CA211052, R01CA204302, U01CA217882 and the Chan-Zuckerberg Biohub (to T.G.B.), Pfizer, as well as the University of California Cancer League (to C.E.M.), AstraZeneca, the Damon Runyon Cancer Research Foundation P0528804, Doris Duke Charitable Foundation P2018110, V Foundation P0530519 and NIH/NCI R01CA227807 (to C.M.B.). F.H. was supported by the Mildred Scheel postdoctoral fellowship from the German Cancer Aid. E.A.Y. is supported by T32 HL007185 from the NHLBI. Cancer studies in the Harris Lab are supported in part by the National Cancer Institute (P01-CA234228). R.S.H. is the Ewing Halsell President's Council Distinguished Chair at the University of Texas San Antonio and an Investigator of the Howard Hughes Medical Institute. D.R.C. was supported by the Francis Crick Institute receives its core funding from Cancer Research UK (FC001169), the UK Medical Research Council (FC002269) and the Wellcome Trust (FC001169), as well as an NC3Rs training fellowship (NC/S001832/1). J.S.R.-F. is funded in part by the Breast Cancer Research Foundation, by a Susan G. Komen Leadership grant and by the NIH/NCI (grant P50 CA247749 01). H.Y. is funded in part by NIH/NCI (grant P50 CAS247749 01) and 1R01CA264078-01. M.J.-H. has received funding from CRUK, NIH National Cancer Institute, International Association for the Study of Lung Cancer (IASLC) International Lung Cancer Foundation, Lung Cancer Research Foundation, Rosetrees Trust, UK and Ireland Neuroendocrine Tumour Society (UKI NETs) and NIHR. Special thanks to the Biological Research Facility at the Francis Crick Institute, specifically to A. Adekoya, J. Cormack, A. Horwood and S. Lighterness for their hard work and support. Special thanks also to the Experimental Histopathology Laboratory at the Francis Crick Institute, specifically to E. Nye, B. Almeida, M. Green and R. Stone for their help and support. Special thanks to all the members of the Bivona Laboratory (former and current), D. Gordenin, A. Sweet-Cordero, S. Bandyopadhyay, M. Breese, S. Kaushik, B. Leonard, S. Raju and K. Descamp for their insights and support and S. Elmes, A. Maynard, D.V. Allegakoen and A. Tambe for their technical support.

## Author contributions

D.R.C., P.G., M.K.M., E.K.L., N.K., R.S.H., J.D., T.G.B. and C.S. conceived and designed the study. D.R.C., P.G., M.K.M., J.B., J.D.D.P.V., F.H., B.G., T.M., W.T., T.A., P.A., S.N., C.G., E.G., M.A.B., A.N., F.G.V., W.H., W.T.L., B.A., M.G., C.M., J.P., E.G., C.Z., S.L., J.C., B.R., W.B., A.R., B.A., R.I.V., M.M., N.J.T., T.J.H., C.E.W., N.K., S.V., K.V., S.H., V.O., D.B., M.T., S.D.C.T., R.V., V.B., X.Z. and Y.J. conducted data acquisition for cell line and animal studies. C.M.R., M.D., M.A., C.B., O.P., B.B., C.E.M., J.R.M.B., C.M.B., D.L.K., J.K.R., A.M., J.R.F., P.S., H.Y., M.J.H., P.A., E.A.Y. and L.T. performed data acquisition for clinical studies. C.B., O.P., B.B., M.D., M.A., N.I.V., N.A.T., W.W., L.C., E.M.V.A., J.Y. and J.B. conducted mutational signature analysis and/or other computational analyses. D.R.C, P.G., M.K.M., N.I.V., T.G.B., C.S., E.K.L, R.S.H., W.L.B., L.K.L., C.D., P.P.A., J.P., T.M., M.A.B., A.N., M.D., C.M.R., S.F.B., S.K.C., S.L.P., A.S.B., N.M., C.M., B.R., B.B., W.W., K.H.V., D.L.K., F.H., C.B., O.P., B.B., N.K., N.A.T., U.G. and N.R. were involved in the analysis and interpretation of data. D.R.C., P.G., M.K.M., M.D., M.A., C.B., O.P., K.H.V., N.M., E.M.V.A., N.K., R.S.H., J.D., T.G.B. and C.S. were responsible for drafting and revising the manuscript.

## Competing interests

T.G.B. is an advisor to Novartis, AstraZeneca, Revolution Medicines, Array/Pfizer, Springworks, Strategia, Relay, Jazz, Rain, Engine, Granule Therapeutics and EcoR1 and receives research funding from Novartis and Revolution Medicines, Kinnate, Verastem and Strategia. N.I.V. served on an advisory board for Sanofi Genzyme. C.S. acknowledges grants from AstraZeneca, Boehringer-Ingelheim, Bristol Myers Squibb, Pfizer, Roche-Ventana, Invitae (previously Archer Dx—collaboration in minimal RD sequencing technologies), Ono Pharmaceutical, and Personalis. He is the chief investigator for the AZ MeRmaiD 1 and 2 clinical trials and is the Steering Committee Chair. He is also co-chief investigator of the NHS Galleri trial funded by GRAIL and a paid member of GRAIL's Scientific Advisory Board (SAB). He receives consultant fees from Achilles Therapeutics (also an SAB member), Bicycle Therapeutics (also an SAB member), Genentech, Medicxi, China Innovation Center of Roche (CICoR) formerly Roche Innovation Center—Shanghai, Metabomed (until July 2022), Relay Therapeutics and the Sarah Cannon Research Institute. C.S. has received honoraria from Amgen, AstraZeneca, Bristol Myers Squibb, GlaxoSmithKline, Illumina, MSD, Novartis, Pfizer and Roche-Ventana; has previously held stock options in Apogen Biotechnologies and GRAIL; currently has stock options in Epic Bioscience and Bicycle Therapeutics and has stock options and is a cofounder of Achilles Therapeutics. C.S. declares a patent application

(PCT/US2017/028013) for methods to lung cancer; targeting neoantigens (PCT/EP2016/059401); identifying patent response to immune checkpoint blockade (PCT/EP2016/071471), determining HLA LOH (PCT/GB2018/052004); predicting survival rates of patients with cancer (PCT/GB2020/050221), identifying patients who respond to cancer treatment (PCT/GB2018/051912); methods for lung cancer detection (US20190106751A1). He is an inventor on a European patent application (PCT/GB2017/053289) relating to assay technology to detect tumor recurrence. This patent has been licensed to a commercial entity, and under their terms of employment, C.S. is due a revenue share of any revenue generated from such license(s). E.M.V.A. is a consultant for Tango Therapeutics, Genome Medical, Invitae, Enara Bio, Janssen, Manifold Bio, Monte Rosa; receives research funding from Novartis, BMS; has equity in Tango Therapeutics, Genome Medical, Syapse, Enara Bio, Manifold Bio, Microsoft and Monte Rosa; has received travel reimbursement from Roche/Genentech and own institutional patents filed on chromatin mutations and immunotherapy response, and methods for clinical interpretation. C.E.M. is on the advisory board of Genentech; receives honoraria from Novartis, Guardant, Research and receives funding from Novartis, Revolution Medicines. C.M.B. is a consultant for Amgen, Foundation Medicine, Blueprint Medicines and Revolution Medicines; receives research funding from Novartis, AstraZeneca and Takeda and receives institutional research funding from Mirati, Spectrum, MedImmune and Roche. J.S.R.-F. reports receiving personal/consultancy fees from Goldman Sachs, Bain Capital, REPARE Therapeutics, Saga Diagnostics and Paige.AI, membership of the SAB of VolitionRx, REPARE Therapeutics and Paige.AI, membership of the Board of Directors (BOD) of Grupo Oncoclinicas, and ad hoc SAB of Astrazeneca, Merck, Daiichi Sankyo, Roche Tissue Diagnostics and Personalis, outside the scope of this study. H.Y. receives consulting fees from AstraZeneca, Daiichi, Taiho, Janssen, AbbVie, Blueprint, Black Diamond Research funding to my institution from AstraZeneca, Daiichi, Cullinan, Janssen, Blueprint, Black Diamond, Novartis, Pfizer, ERASCA. S.F.B. owns equity in, receives compensation from, serves as a consultant for and serves on the SAB and BOD of Volastra Therapeutics. He serves on the scientific advisory board of Meliora Therapeutics. M.J.-H. has consulted for, and is a member of, the Achilles Therapeutics Scientific Advisory Board and Steering Committee; has received speaker honoraria from Pfizer, Astex Pharmaceuticals, Oslo Cancer Cluster and Bristol Myers Squibb and is listed as a co-inventor on a European patent application relating to methods to detect lung cancer (PCT/US2017/028013). This patent has been licensed to commercial entities and, under terms of employment, M.J.-H. is due a share of any revenue generated from such license(s). The other authors have no competing interests to declare.

## Additional information

**Extended data** is available for this paper at https://doi.org/10.1038/s41588-023-01592-8.

**Correspondence and requests for materials** should be addressed to Deborah R. Caswell or Trever G. Bivona.

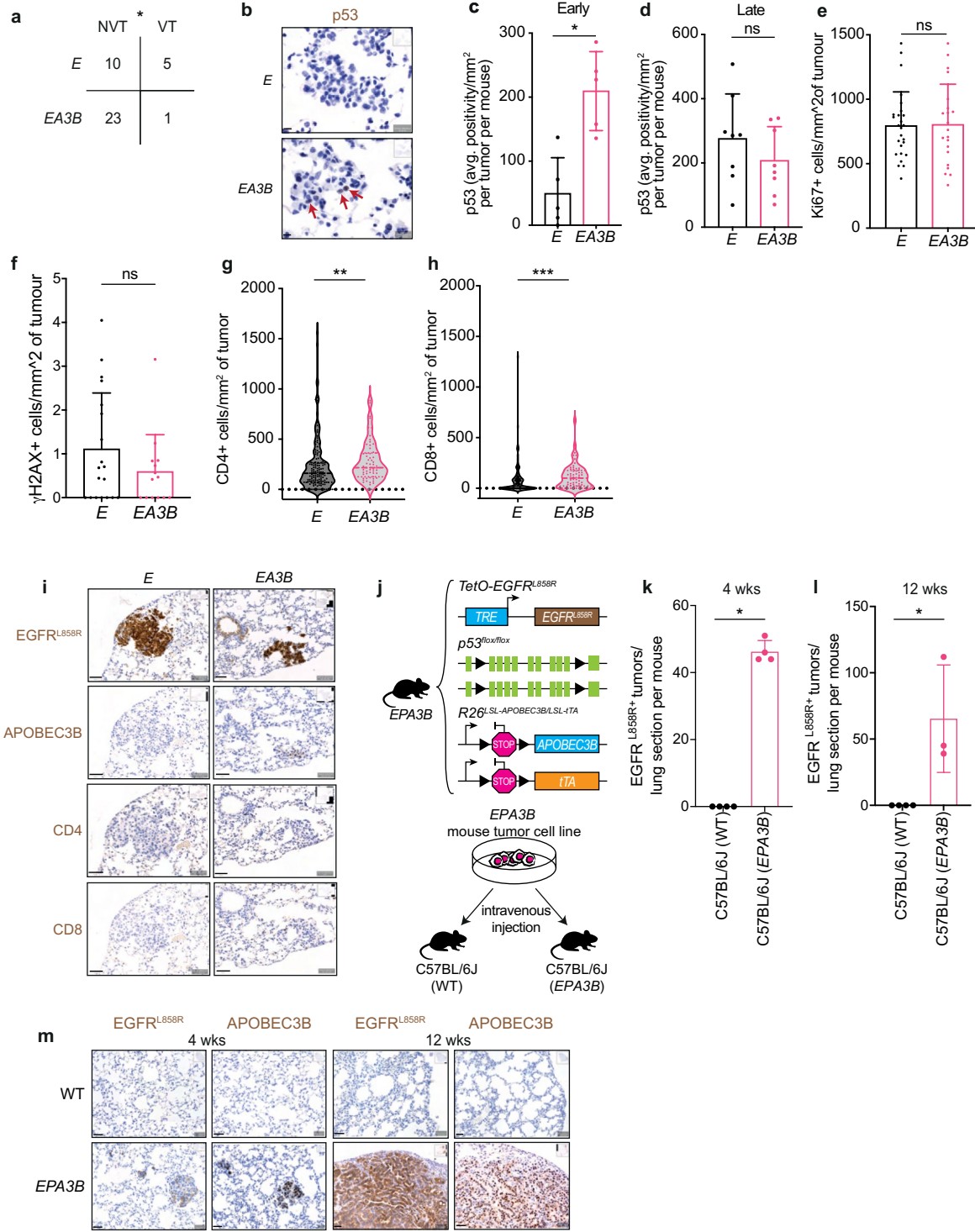

**Extended Data Fig. 1 | See next page for caption.**

**Extended Data Fig. 1 | APOBEC3B is detrimental for tumorigenesis in an *EA3B* mouse model of lung cancer. a**, Two by two contingency table of the number of mice with visible tumors (VT) or no visible tumors (NVT) by microCT at 3 months (two-sided Fisher's exact test, *P = 0.0236). **b**, Representative images of p53 nuclear IHC staining (scale bar=10 μm, arrows indicate positive cells, *E* n = 5, *EA3B* n = 5 biological replicates). **c**, Quantification of p53 positive cells per lung area by IHC staining at 3 months post-induction (*E* n = 5, *EA3B* n = 5, mean ± SD, two-sided Mann-Whitney test, *P = 0.0159). **d**, Quantification of p53 positive cells per lung area by IHC staining at late timepoint (termination) (*E* n = 8, *EA3B* n = 8, mean ± SD, two-sided Mann-Whitney test). **e**, Quantification of Ki67-positive cells per mm² of tumor at 3 months post-induction (*E* n = 9, *EA3B* n = 10, each dot represents a tumor, mean ± SD, two-sided unpaired t-test). **f**, Quantification of γH2AX-positive cells per mm² of tumor at 3 months post-induction (E n = 9, EA3B n = 10, each dot represents a tumor, mean ± SD, two-sided Mann-Whitney test). **g**, Quantification of CD4+ cells per mm² of tumor at 3 months post-induction (*E* n = 8, *EA3B* n = 7, each dot represents a tumor, mean ± SD, two-sided Mann-Whitney test, **P = 0.0086). **h**, Quantification of CD8+ cells per mm² of tumor at

3 months post-induction (*E* = 8, *EA3B* = 8, each dot represents a tumor, mean ± SD, two-sided Mann-Whitney test, ***P = 0.0003). **i**, Representative IHC stainings of EGFR^L858R, APOBEC3B, and CD4 and CD8 T cells (scale bar=50 μm, EGFR^L858R *E* n = 9, *EA3B* n = 10, A3B *E* n = 9, *EA3B* n = 10, p53^fl/fl *E* n = 5, *EA3B* n = 5, CD4 *E* n = 8, *EA3B* n = 7, CD8 *E* n = 8, *EA3B* n = 8). **j**, Intravenous transplantation using an *EGFR^L858R; p53fl/fl;APOBEC3B (EPA3B)* mouse tumor cell line injected into a wildtype C57BL/6J mouse or a C57BL/6J *EPA3B* GEMM mouse. **k**, Quantification of EGFR^L858R positive tumors in C57BL/6 wildtype versus *EPA3B* mice at 4 weeks (mean ± SD, two-sided Mann-Whitney test, n = 4, *P = 0.0286, each dot represents a mouse, C57BL/6 wildtype n = 4, C57BL/6J *EPA3B* GEMM n = 4). **l**, Quantification of EGFR^L858R positive tumors in C57BL/6 wildtype versus *EPA3B* mice at 12 weeks (mean ± SD, two-sided Mann-Whitney test, n = 3, *P = 0.0286, each dot represents a mouse, C57BL/6 wildtype n = 4, C57BL/6J *EPA3B* GEMM n = 3). **m**, Representative IHC staining of EGFRL858R and APOBEC3B (scale bar=50 μm, 4 weeks C57BL/6 wildtype n = 4, C57BL/6J *EPA3B* GEMM n = 4, 12 weeks C57BL/6 wildtype n = 4, C57BL/6J *EPA3B* GEMM n = 3).

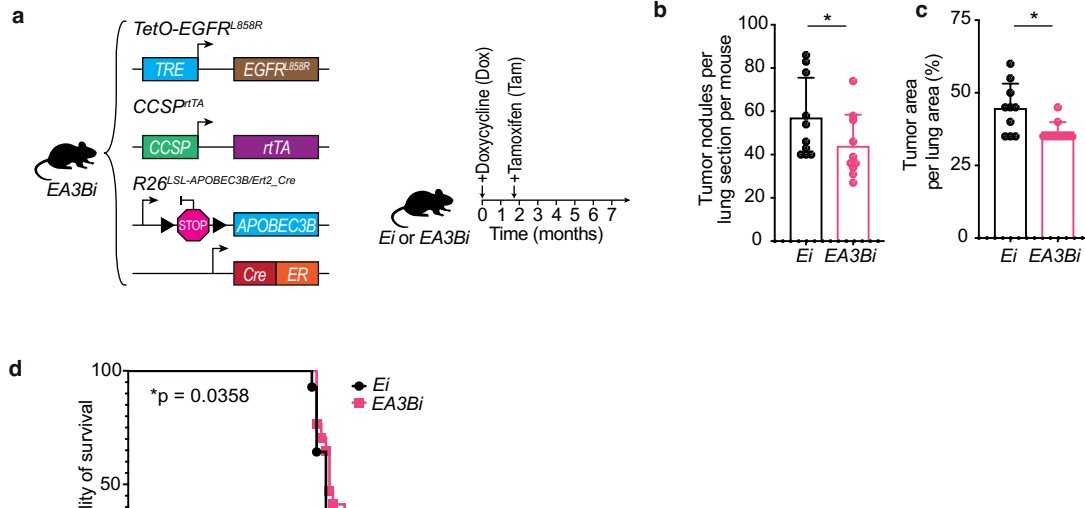

**Extended Data Fig. 2 | Subclonal A3B expression in treatment naive mice inhibits tumor growth. a**, Experimental set up of induction of subclonal *APOBEC3B* using *TetO-EGFR^L858R^;CCSP-rtTA;Rosa26^LSL-APOBEC3B/Cre-ER(T2)^(EA3Bi)* or *TetO-EGFR^L858R^;CCSP-rtTA;Rosa26^Cre-ER(T2)/+^(Ei)* mice. **b**, Tumor nodules per lung section per mouse at termination (*Ei* n = 10, *EA3Bi* n = 10, two-sided Mann-Whitney test,

*P = 0.0494). **c**, Tumor area per lung area at termination (*Ei* n = 10, *EA3Bi* n = 10, two-sided Mann-Whitney test, *P = 0.0216). **d**, Survival curve of *Ei* versus *EA3Bi* mice (*Ei* n = 14, *EA3Bi* n = 17, each dot represents a mouse, Log-rank (Mantel-Cox) test, *P = 0.0358).

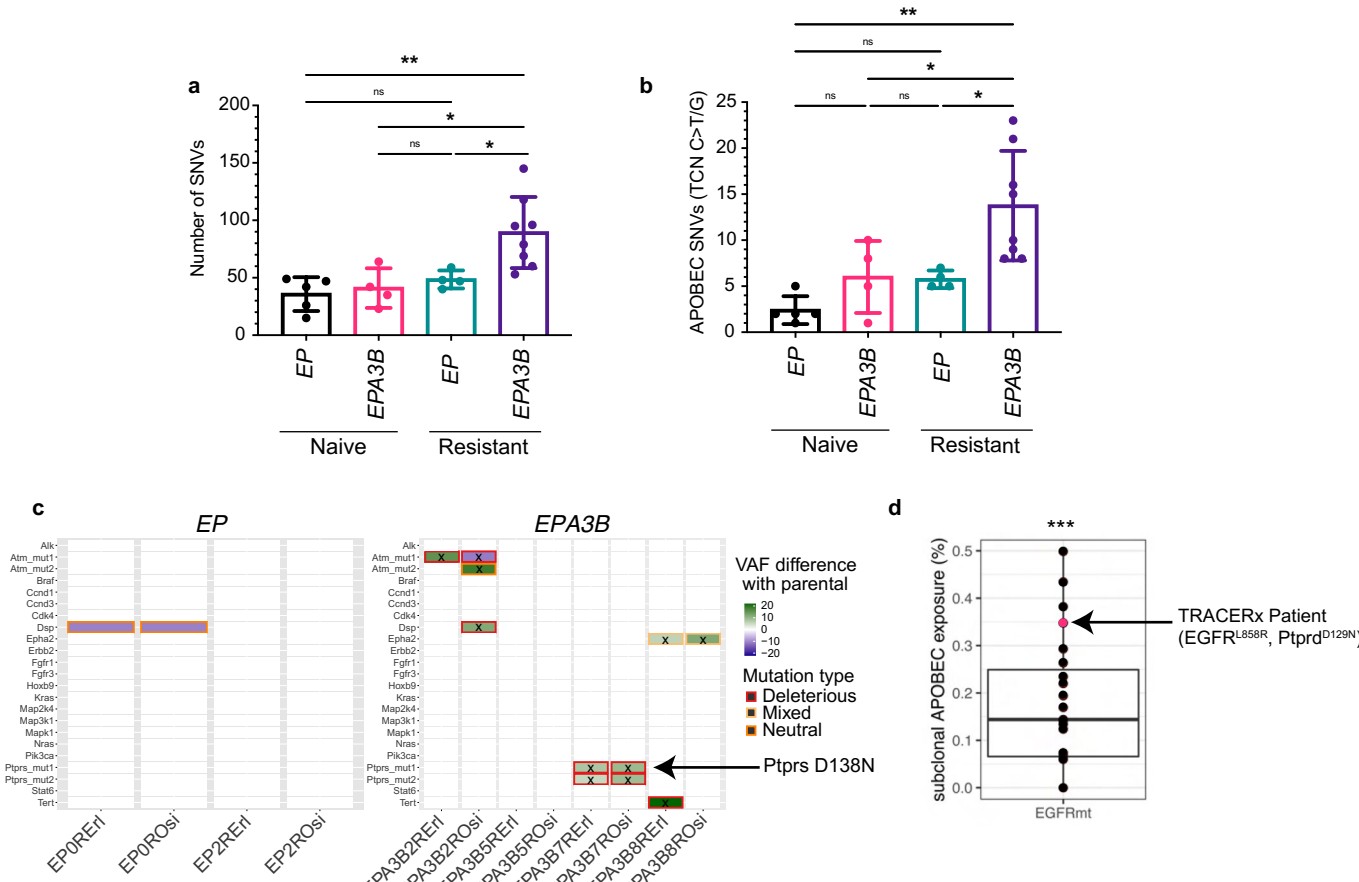

**Extended Data Fig. 3 | Putative resistance mutations in genes previously associated with TKI resistance in mouse tumor cell lines. a**, Comparison of *EP* and *EPA3B* mutation burdens in TKI naive and TKI resistant mouse lung cancer cell lines (mean ± SD, one-way ANOVA test, *P = 0.0135, *P = 0.0346, **P = 0.0039). **b**, Comparison of *EP* and *EPA3B* APOBEC driven mutations (TCN, C > T or C > G SNVs) in TKI naive and TKI resistant mouse lung cancer cell lines (mean ± SD, one-way ANOVA test, *P = 0.0333, *P = 0.0333, **P = 0.0012). **c**, Functional annotation of TCN mutations in potential TKI resistance genes with change in variant allele frequency shown (x=TCN, Red square=deleterious mutation, yellow square=mixed (neutral and deleterious), orange square=neutral). **d**, Significant subclonal enrichment of the APOBEC-associated mutation signature in the TRACERx patient with A3B driven D129N mutation in the type IIa PTP PTPRD (equivalent to D138N mutation in PTPRS ***P = 0.0002, two-sided one-sample Wilcoxon test).

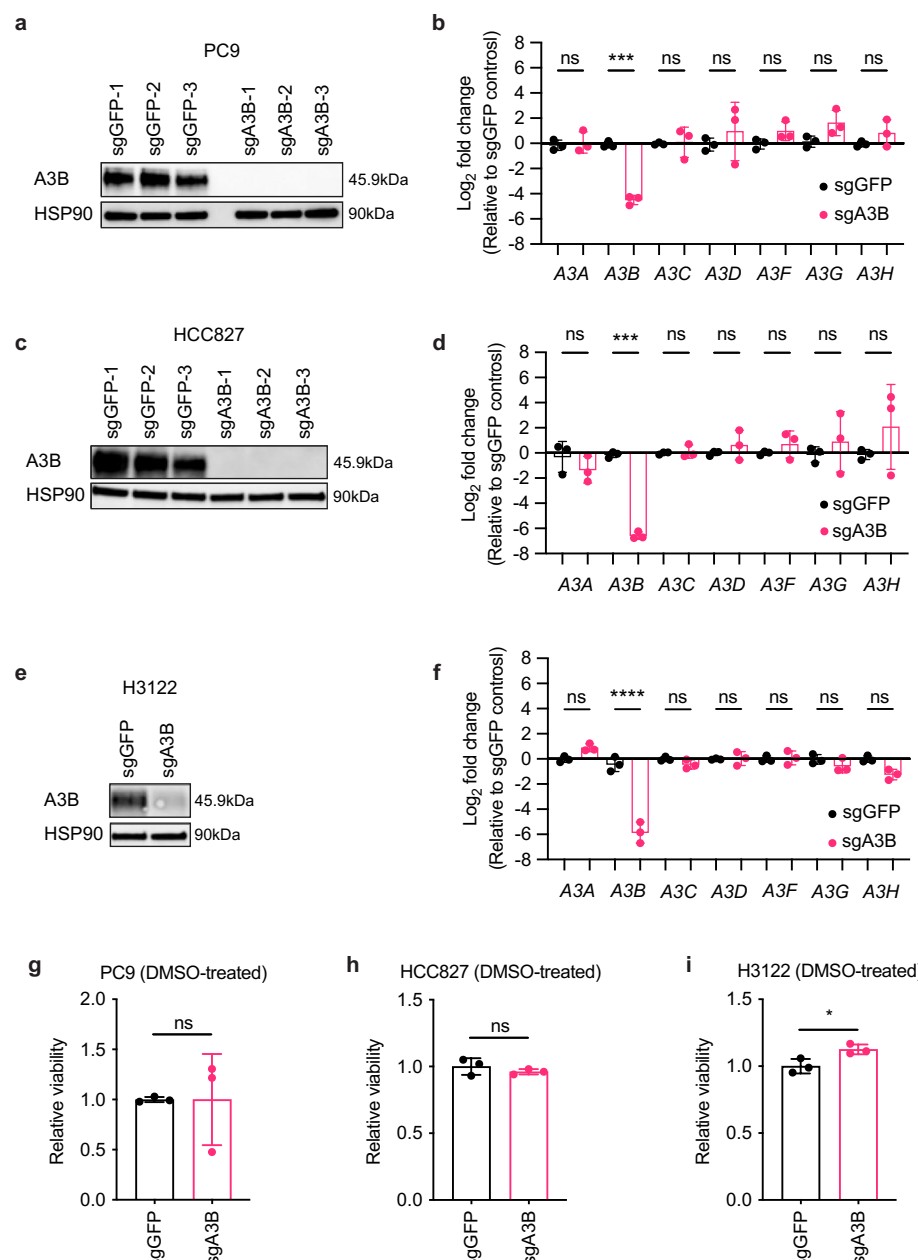

**Extended Data Fig. 4 | APOBEC3 family member mRNA and protein levels in control and A3B knockout cell lines. a**, Immunoblot for APOBEC3B (A3B) protein levels in PC9 control (sgGFP) and A3B knockout (sgA3B) cell lines, (n = 3 biological replicates, 2 independent experiments). **b**, mRNA expression levels of *APOBEC3* family members in control (sgGFP) and *A3B* knockout (sgA3B) PC9 cell lines (n = 3 biological replicates, mean ± SD, one-way ANOVA test, ***P = 0.0001). **c**, Immunoblot for A3B protein levels in HCC827 control (sgGFP) and *A3B* knockout (sgA3B) cell lines (n = 3 biological replicates, 2 independent experiments). **d**, mRNA expression levels of *APOBEC3* family members in control (sgGFP) and *A3B* knockout (sgA3B) HCC827 cell lines (n = 3 biological replicates, mean ± SD, one-way ANOVA test, ***P = 0.0001). **e**, Immunoblot for A3B protein

levels in H3122 control (sgCtrl) or *A3B* knockout (sgA3B) cell line (n = 1 biological replicate, 2 independent experiments). **f**, mRNA expression levels of *APOBEC3* family members in control (sgGFP) and *A3B* knockout (sgA3B) H3122 cell lines (n = 2 biological replicates, mean ± SD, one-way ANOVA test, ****P < 0.0001). **g**, CellTiter-Glo (CTG) viability assay performed on *A3B*-deficient or *A3B*-proficient PC9 cells treated with DMSO for 7 days (n = 3 biological replicates, mean ± SD, two-sided t-test). **h**, CTG viability assay performed on *A3B*-deficient or *A3B*-proficient HCC827 cells treated with DMSO for 7 days (n = 3 biological replicates, mean ± SD, two-sided t-test). **i**, CTG viability assay performed on *A3B*-deficient or *A3B*-proficient H3122 cells treated with DMSO for 7 days (n = 3 biological replicates, mean ± SD, two-sided t-test, *P = 0.0293).

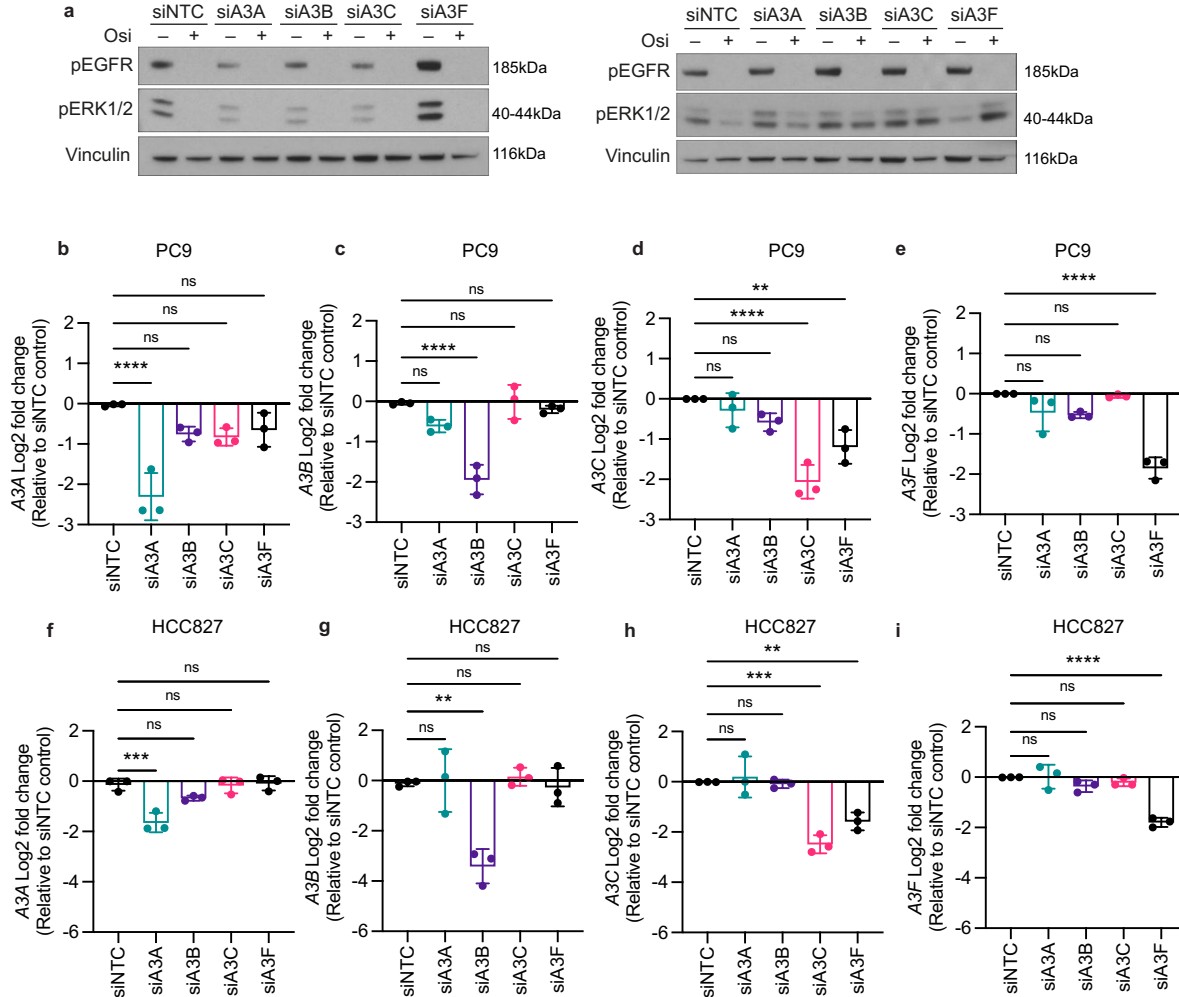

**Extended Data Fig. 5 | Knockdown of APOBEC3 family members under TKI treatment. a**, Western blot analyses for pEGFR and pERK1/2 to confirm loss with osimertinib treatment in PC9 and HCC827 cells treated with DMSO or 0.5 μM osimertinib (Osi) for 18 hours (PC9 n = 4 independent experiments, HCC827 n = 1 independent experiment). **b–e**, RT-qPCR analysis of *APOBEC3* family members expression in PC9 cells treated with DMSO or 0.5 μM osimertinib for 18 hours, with siRNA knockdown of *APOBEC3A (A3A), APOBEC3B (A3B), APOBEC3C (A3C) or APOBEC3F (A3F)*: *A3A* expression (**b**, n = 3 biological replicates, mean ± SD, one-way ANOVA test ****P < 0.0001); *A3B* expression (**c**, n = 3 biological replicates, mean ± SD, one-way ANOVA test, ****P < 0.001); *A3C* expression (**d**, n = 3 biological replicates, mean ± SD, one-way ANOVA test, **P = 0.0049, ****P < 0.0001); *A3F* expression (**e**, n = 3 biological replicates, mean ± SD, one-way ANOVA test ****P = < 0.001). **f–i**, RT-qPCR analysis of *APOBEC3* family members expression in HCC827 cells treated with DMSO or 0.5 μM osimertinib for 18 hours, with siRNA knockdown of *A3A, A3B, A3C or A3F*: *A3A* expression (**f**, n = 3 biological replicates, mean ± SD, one-way ANOVA test, ***P = 0.0003); *A3B* expression (**g**, n = 3 biological replicates, mean ± SD, one-way ANOVA test, **P = 0.0011); *A3C* expression (**h**, n = 3 biological replicates, mean ± SD, one-way ANOVA test, ***P = 0.0002, **P = 0.0040); *A3F* expression (**i**, n = 3 biological replicates, mean ± SD, one-way ANOVA test, ****P < 0.0001).

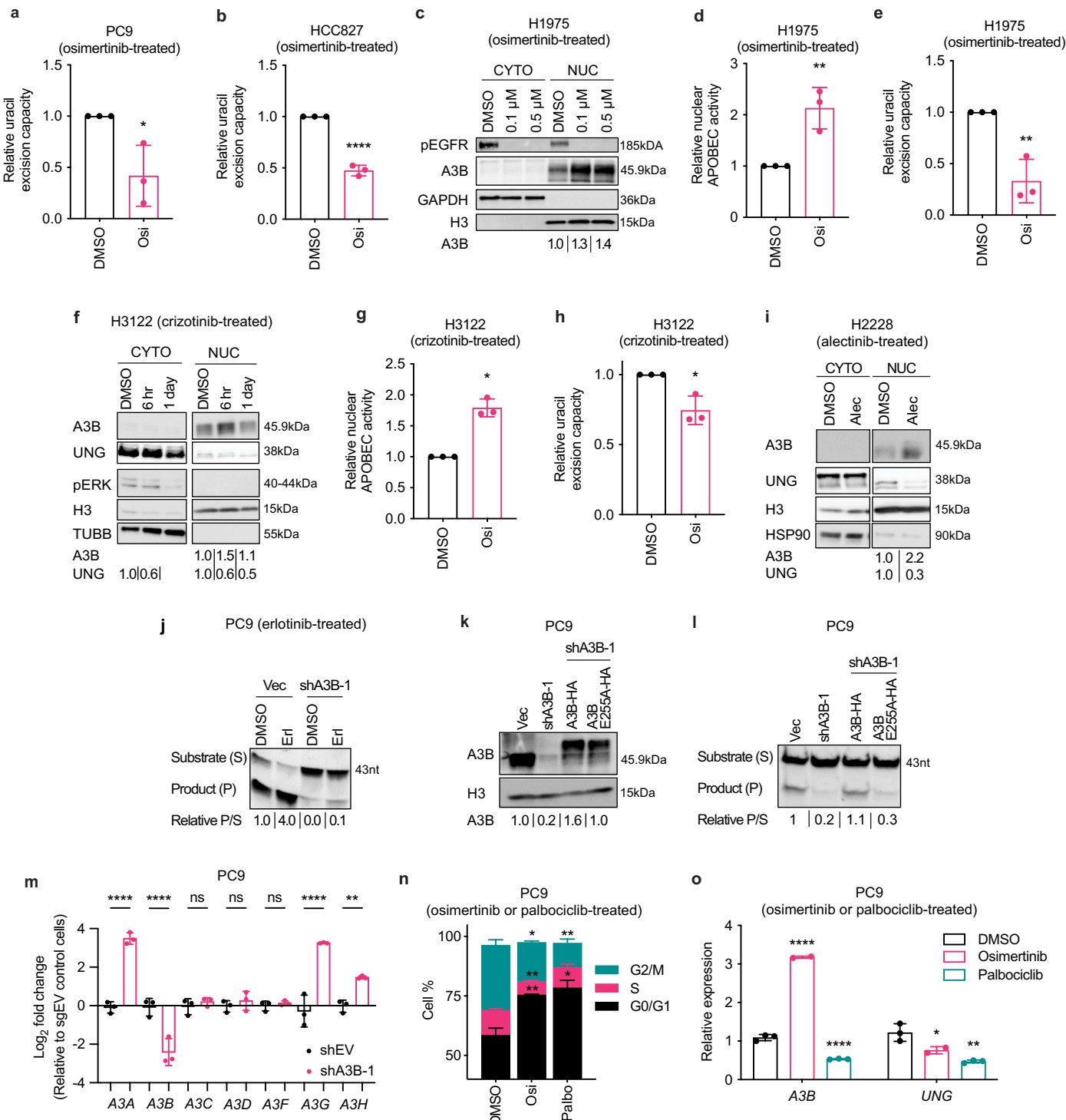

**Extended Data Fig. 6 | See next page for caption.**

**Extended Data Fig. 6 | TKI treatment induces increased A3B and decreased UNG expression and activity in pre-clinical models of lung adenocarcinoma.** **a**, Uracil excision capacity assay (UEC) using PC9 nuclear extracts treated with DMSO or 2 µM osimertinib (Osi) (n = 3 biological replicates, mean ± SD, two-tailed t-test, *P = 0.0275). **b**, UEC in HCC827 treated with DMSO or 0.4 µM osi (n = 3 biological replicates, mean ± SD, two-tailed t-test, ****P < 0.0001). **c**, Western blot (WB) from H1975 treated with DMSO, 0.1 µM or 0.5 µM crizotinib (CYTO: cytoplasmic; NUC: nuclear; H3: Histone H3; TUBB: beta-tubulin) (n = 3 biological replicates). **d**, APOBEC activity assay (AAA) using H1975 treated with DMSO or 1 µM osi (n = 3 biological replicates, mean ± SD, two-tailed t-test, **P = 0.0084). **e**, UEC in H1975 treated with DMSO or 1 uM osi (n = 3 biological replicates, mean ± SD, two-tailed t-test, **P = 0.0054). **f**, WB from H3122 treated with DMSO or 1 µM crizotinib (n = 3 biological replicates). **g**, AAA from H3122 treated with DMSO or 0.5 µM crizotinib (n = 3 biological replicates, mean ± SD, two-tailed t-test, *P = 0.0204). **h**, UEC in H3122 treated with DMSO or 0.5 µM crizotinib (n = 3 biological replicates, mean ± SD, two-tailed t-test, *P = 0.0123).

**i**, WB of H2228 treated with DMSO or 0.5 µM alectinib for (n = 3 biological replicates). **j**, AAA from PC9 transduced with empty vector (shEV) or shRNA against A3B (shA3B-1) and treated with DMSO or 1 µM erlotinib (n = 3 biological replicates). **k**, WB from nuclear extracts of PC9 transduced with shEV or shA3B-1 alone or together with wild-type HA-tagged A3B or HA-tagged catalyticaly-inactive A3B mutant (E255A) expression plasmid (n = 3 biological replicates). **l**, AAA from PC9 as in panel **k**, in the absence of RNase A (n = 3 biological replicates). **m**, mRNA expression levels of *APOBEC3* family members in control (shEV) and *A3B* knockdown (shA3B) PC9 (n = 3 biological replicates, mean ± SD, one-way ANOVA test, **P = 0.0059, ****P < 0.0001). **n**, Cell cycle analysis of PC9 treated with DMSO, 2 µM osimertinib or 1 µM palbociclib (Palbo) (n = 4 biological replicates, mean ± SD, two-tailed t-tests, *P = 0.012, **P = 0.0032, **P = 0.0071, **P = 0.0084, *P = 0.0105). **o**, RT-qPCR analysis of PC9 cells treated as in panel **a**, (n = 2 or 3 biological replicates, mean ± SD, one-way ANOVA test, ****P < 0.0001, *P = 0.0215, **P = 0.0018). Panels **a–i**, n: treatment for 18 hours.

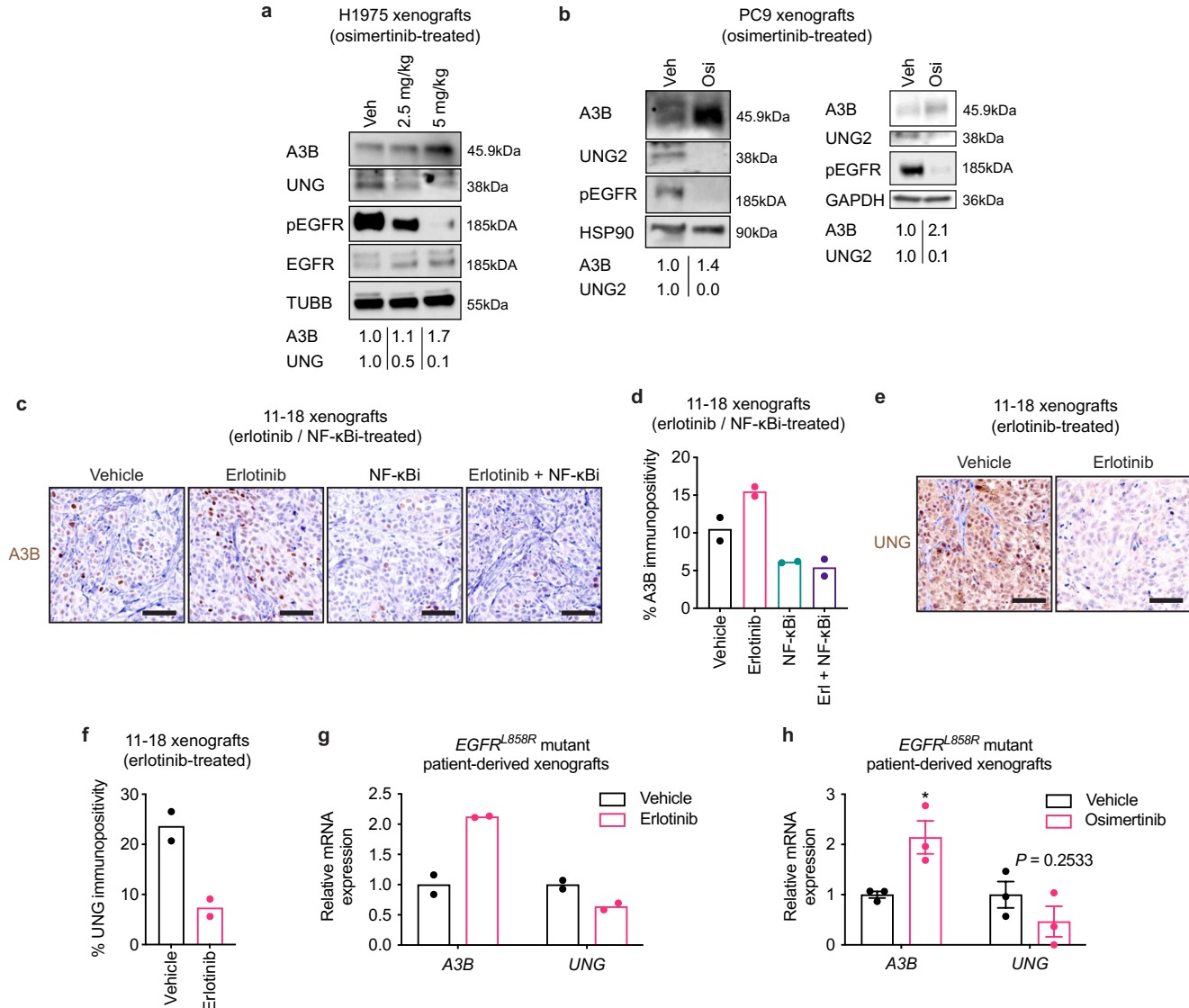

**Extended Data Fig. 7 | EGFR inhibition induces A3B upregulation and UNG downregulation in xenograft models. a**, Western blot analysis using extracts of EGFR-mutant H1975 human NSCLC xenografts harvested after 4 days of treatment with vehicle or the indicated doses of osimertinib (TUBB: Tubulin Beta Class I) (n = 1 biological replicate). **b**, Western blot analyses of extracts of PC9 tumor xenografts treated with vehicle or 5 mg/kg osimertinib (n = 2 biological replicates). **c**, Representative images of IHC analysis of APOBEC3B (A3B) protein levels in 11-18 xenografts treated with vehicle, 12.5 mg/kg/day erlotinib, 7.5 mg/kg/day NF-κB inhibitor (NF-κBi, PBS-1086) or combination (Erlotinib + NF-κBi) for 2 months (scale: 60 μM, n = 2 biological replicates)17. **d**, Quantification of immunohistochemical staining for A3B in 11-18 xenografts treated with vehicle, erlotinib (Erl), NF-κB inhibitor (NF-κBi, PBS-1086) or combination (Erl + NF-κBi) for 2 months (n = 2 biological replicates). **e**, Representative images of IHC analysis of UNG protein levels in 11-18 xenografts treated with vehicle or 12.5 mg/kg/day erlotinib for 2 months (n = 2 biological replicates). **f**, Quantification of immunohistochemical staining for UNG in 11-18 xenografts treated with vehicle or erlotinib for 2 months (n = 2 biological replicates). **g**, RNA-Seq analysis upon treatment of a PDX model of human EGFR-driven lung adenocarcinoma with vehicle or erlotinib (2 days, 25 mg/kg) (n = 2 biological replicates). **h**, RNA-Seq analysis upon treatment of a PDX model of human EGFR-driven lung adenocarcinoma with vehicle or osimertinib (6 days, 10 mg/kg) (n = 3 biological replicates, mean ± SD, two-sided t-test, *P = 0.0267).

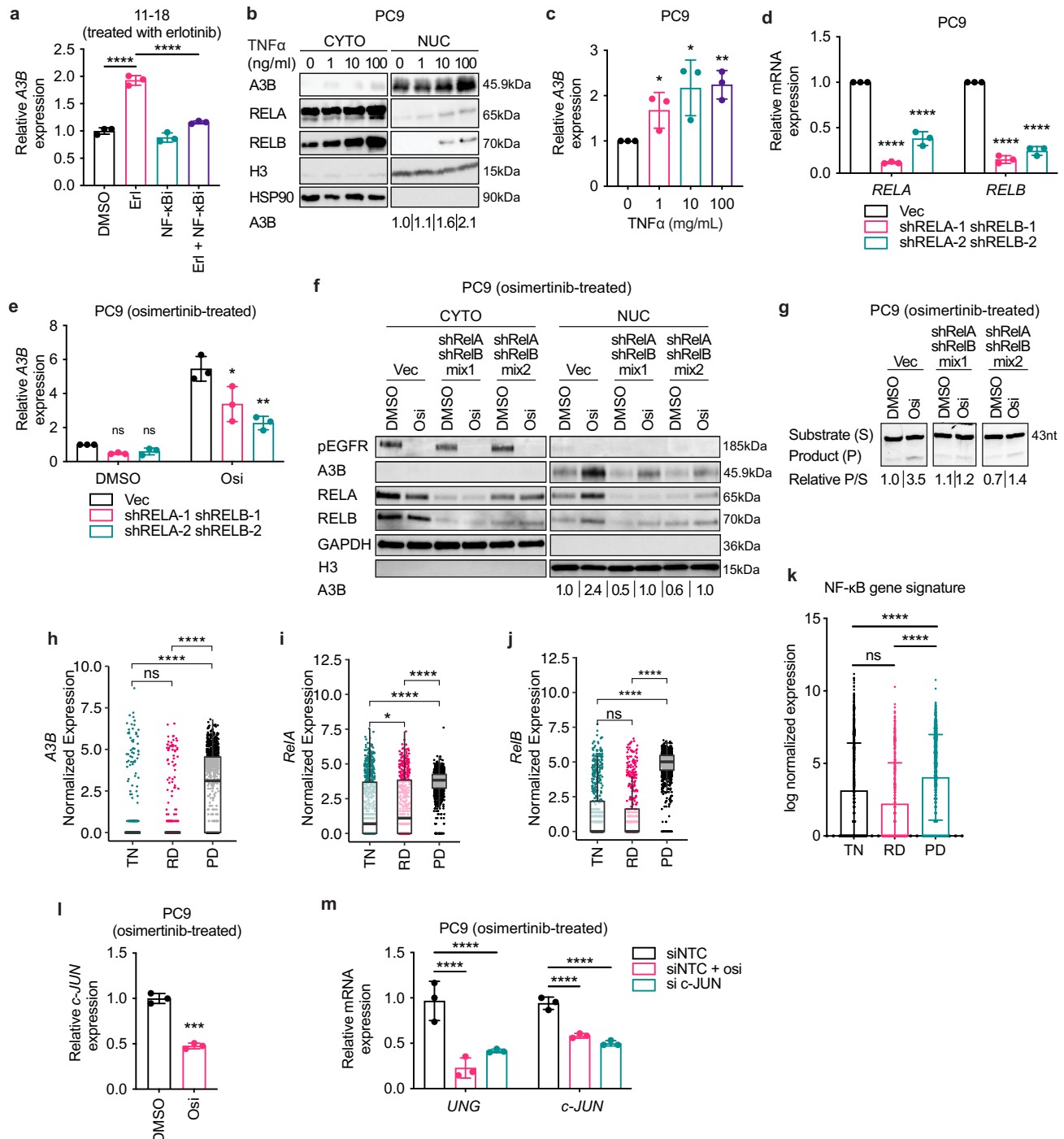

**Extended Data Fig. 8 | NF-κB signaling contributes to TKI-induced A3B upregulation, and expression of c-Jun and UNG are decreased upon TKI treatment. a**, RNA-Seq analysis of EGFR-mutant 11-18 cells treated with DMSO, 100 μM erlotinib (erl), 5 μM NF-κB inhibitor (NF-κBi, PBS-1086) or combination (Erl+NF-κBi) (n = 3 biological replicates, mean ± SEM, one-way ANOVA test, ****P < 0.0001). **b**, Western blot analysis of extracts from PC9 treated with DMSO or with TNFα for 8.5 hours (n = 3 biological replicates). **c**, RT-qPCR analysis of TNFα-treated PC9 (n = 3 biological replicates, mean ± SD, two-tailed t-test, *P = 0.0406, *P = 0.0299, **P = 0.0024). **d**, RT-qPCR validation of *RELA* and *RELB* knockdown in PC9 with non-targeting vector or combination of shRELA-1+shRELB-1 (mix1) or shRELA-2+shRELB-2 (mix2) (n = 3 biological replicates; mean ± SD, one-way ANOVA test, ****P < 0.0001). **e**, RT-qPCR analysis of *APOBEC3B* (*A3B*) in PC9 with non-targeting vector or mix1 or mix2, treated with DMSO or 500 nM osi for 1 day (n = 3 biological replicates; mean ± SD, two-tailed t-test, *P = 0.0465, **P = 0.0026). **f**, Western blot analysis of PC9 used in e (n = 3

biological replicates). **g**, APOBEC activity assay of PC9 used in f (n = 3 biological replicates). **h–j**, Single-cell RNA-Seq expression in lung cancer cells from patient tumors at treatment naïve (TN, 762 cells), residual disease (RD, 553 cells) and progressive disease (PD, 988 cells) of: *A3B* (**h**), *RelA* (**i**) and *RelB* (**j**) (all data points shown, two-sided Wilcoxon test with Holm correction, ****P < 2.22e-16). **k**, Single-cell RNA-Seq analysis of NF-κB signature (from Gilmore_Core_NFκB_Pathway, GSEA, C2) in tumors from panels **h–j** (mean ± SD, two-sided Wilcoxon test with Holm correction, ****P < 2.22e-16). **l**, RT-qPCR analysis of *c-JUN* in PC9 treated with DMSO or 2 μM osimertinib for 9 days (n = 3 biological replicates, mean ± SEM, two-tailed t-test, ***P = 0.0009). **m**, RT-qPCR analysis of PC9 with non-targeting (siNTC) or *c-JUN* siRNA, treated with DMSO or 2 μM osimertinib for 18 hours (n = 3 biological replicates, mean ± SD, one-way ANOVA test, ****P < 0.0001). Boxplots: middle line=median, lower and upper hinges=first and third quartiles, lower and upper whiskers=smallest and largest values within 1.5×inter-quartile range from hinges.

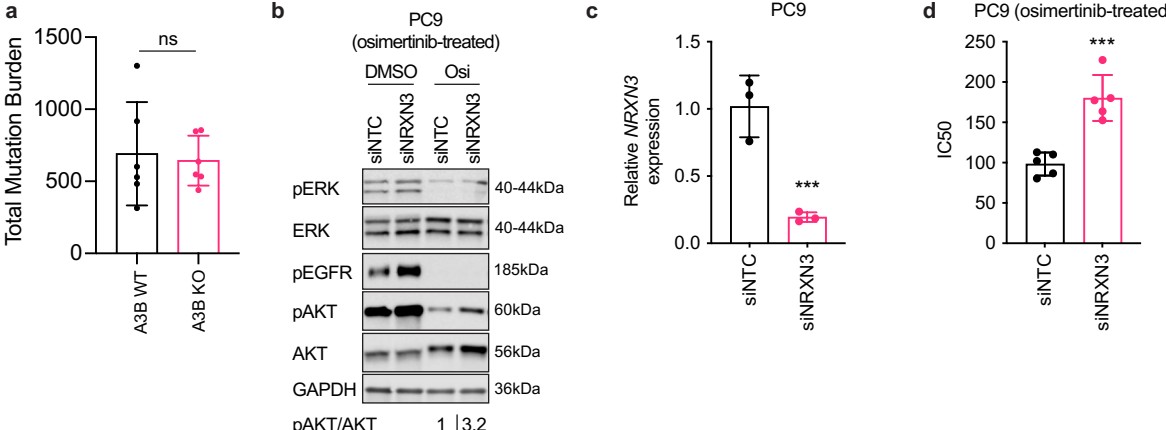

**Extended Data Fig. 9 | Mutation burden and putative resistance mutations in genes previously associated with TKI resistance in PC9 TKI resistant cell line. a**, Mutation burden quantified in APOBEC3B (A3B)-deficient (A3B KO), and A3B-proficient (A3B WT) single cell cloned PC9 cells treated with osimertinib for 3 months (n = 6 biological replicates, mean ± SD, two-tailed Mann-Whitney test). **b**, Western blot analysis of PC9 cells treated with non-targeting (siNTC) or *NRXN3*-targeting (siNRXN3) siRNA and treated with DMSO or 500 nM osimertinib for 2 days (n = 3 biological replicates). **c**, RT-qPCR-based validation of *NRXN3* knockdown in cells shown in a (n = 3 technical replicates, mean ± SD, two-sided t-test performed on ΔCt values shown, ***P = 0.0007). **d**, IC50 analysis of PC9 siNTC or siNRXN3 after 3-day treatment (n = 5 biological replicates for each of the following doses of osimertinib: 0 nM, 5 nM, 50 nM, 100 nM, 500 nM and 5000 nM, mean ± SD, two-sided t-test, ***P = 0.0004).

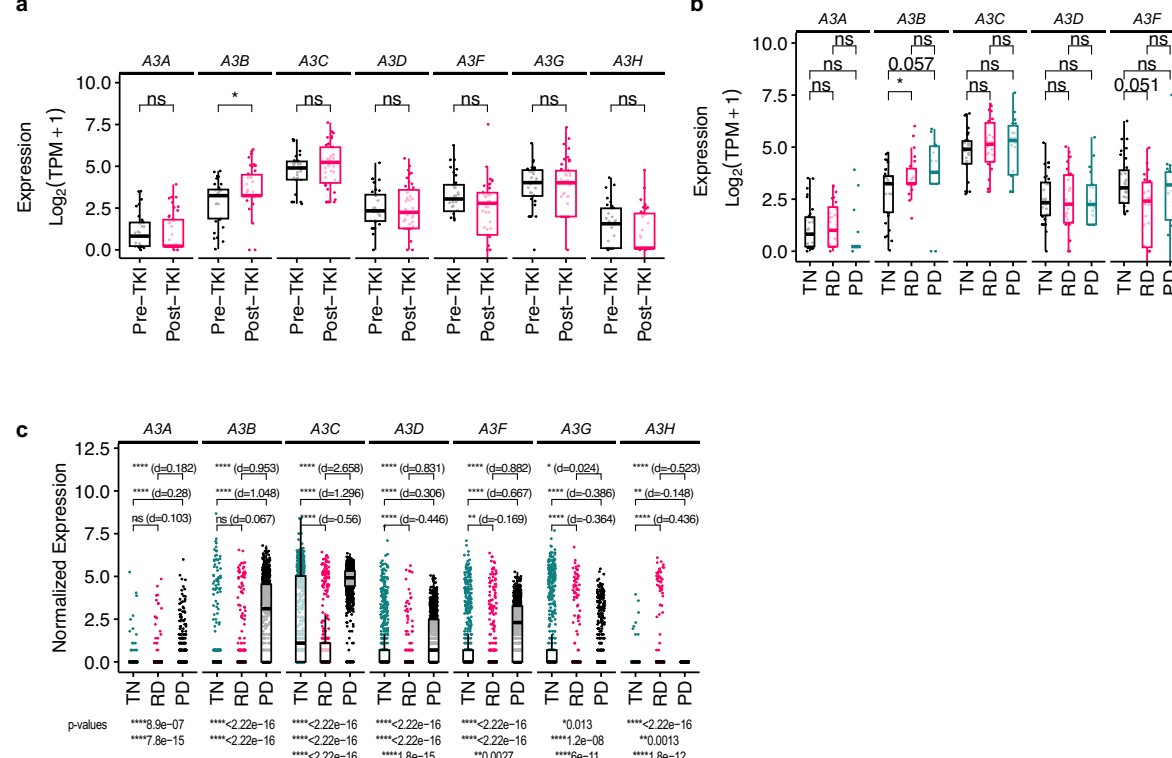

**Extended Data Fig. 10 | Expression of APOBEC3 enzymes in clinical samples upon targeted therapy treatment. a**, Comparison of *APOBEC3B* (*A3B*) expression levels (Exp: batch corrected TPM) measured using RNA-Seq analysis in human NSCLC specimens driven by EGFR and ALK driver mutations obtained before treatment (Pre-TKI, 32 samples), or post-treatment (Post-TKI, 42 samples) (all data points shown, two-sided t-test, *P = 0.011). **b**, Comparison of *APOBEC3* (*A3*) family member expression levels (Exp: batch corrected Log (TPM + 1) measured using RNA-seq analysis in human NSCLC specimens obtained at treatment naïve (TN), residual disease (RD) or progressive disease (PD) with TKI (all data points shown, 762, 553, and 988 cells per group respectively, two-sided Wilcoxon test with Holm correction, *P = 0.02). **c**, Boxplot of normalized *A3* family member expression measured using scRNA-seq obtained from the same samples as b (all data points shown, 762, 553, and 988 cells per group respectively, two-sided Wilcoxon test with Holm correction, *P < 0.05, **P < 0.01, ****P < 0.001, d=effect size calculated using a Cohen test). Boxplots: middle line=median, lower and upper hinges=first and third quartiles, lower and upper whiskers=smallest and largest values within 1.5×inter-quartile range from hinges.

# Reporting Summary

## Statistics

For all statistical analyses, confirm that the following items are present in the figure legend, table legend, main text, or Methods section.

| n/a | Confirmed | |
|---|---|---|
| ☐ | ☒ | The exact sample size (*n*) for each experimental group/condition, given as a discrete number and unit of measurement |
| ☐ | ☒ | A statement on whether measurements were taken from distinct samples or whether the same sample was measured repeatedly |
| ☐ | ☒ | The statistical test(s) used AND whether they are one- or two-sided<br>*Only common tests should be described solely by name; describe more complex techniques in the Methods section.* |
| ☐ | ☒ | A description of all covariates tested |
| ☐ | ☒ | A description of any assumptions or corrections, such as tests of normality and adjustment for multiple comparisons |
| ☐ | ☒ | A full description of the statistical parameters including central tendency (e.g. means) or other basic estimates (e.g. regression coefficient) AND variation (e.g. standard deviation) or associated estimates of uncertainty (e.g. confidence intervals) |
| ☐ | ☒ | For null hypothesis testing, the test statistic (e.g. *F*, *t*, *r*) with confidence intervals, effect sizes, degrees of freedom and *P* value noted<br>*Give P values as exact values whenever suitable.* |
| ☒ | ☐ | For Bayesian analysis, information on the choice of priors and Markov chain Monte Carlo settings |
| ☒ | ☐ | For hierarchical and complex designs, identification of the appropriate level for tests and full reporting of outcomes |
| ☐ | ☒ | Estimates of effect sizes (e.g. Cohen's *d*, Pearson's *r*), indicating how they were calculated |

*Our web collection on statistics for biologists contains articles on many of the points above.*

## Software and code

Policy information about availability of computer code

| Data collection | GraphPad Prism 7, QuantStudio 12K Flex Software V1.3, Adobe Illustrator 26.2.1, ImageJ-2, Microsoft Excel, AnalyzeDirect, QuPath v0.1.2, RSEM (version 1.2.29), VarScan2(v2.4.1), MuTect(v1.1.7) and Scalpel(v0.5.4), PROVEAN (Protein Variation Effect Analyzer), GATK bundle (v2.8), GATK3 (version 3.6.0),  R (version 3.3.1) and deepSNV (v1.18.1), Burrows-Wheeler Aligner (version 0.7.17), Picard (version 2.18.16), deconstructSigs R, MuTect (v1.6), Oncotator, Strelka (v1.0.11), deTiN, DESeq, GSEA. |
|---|---|
| Data analysis | Cell line Whole Genome Mutational Signature Analysis:<br>Sequences were aligned to the human genome (hg38) using the Burrows-Wheeler Aligner (version 0.7.17). PCR duplicates were removed using Picard (version 2.18.16). Reads were locally realigned around InDels using GATK3 (version 3.6.0) tools RealignerTargetCreator to create intervals, followed by IndelRealigner on the aligned bam files. MuTect2 from GATK3 (version 3.6.0) was used in tumour/normal mode to call mutations in test vs control cell lines.  Single nucleotide variants (SNVs) that passed the internal GATK3 filter with read depths over 30 reads at called positions, at least 4 reads in the alternate mutation call and an allele frequency greater than 0.1 were used for downstream analysis. Figures were plotted using the deconstructSigs R package (Rosenthal et al. 2016).<br><br>Mutation analysis:<br>Paired-end reads were aligned to the hg19 human genome using the Picard pipeline (https://gatk.broadinstitute.org/). A modified version of the Broad Institute Getz Lab CGA WES Characterization pipeline (https-docs-google-com.ezp-prod1.hul.harvard.edu/document/d/1VO2kX_fgfUd0x3mBS9NjLUWGZu794WbTepBel3cBg08) was used to call, filter and annotate somatic mutations. Specifically, single-nucleotide variants (SNVs) and other substitutions were called with MuTect (v1.6) (Cibulskis et al., 2013). Mutations were annotated using Oncotator (Ramos et al., 2015). MuTect mutation calls were filtered for 8-OxoG artifacts, and artifacts introduced through the formalin fixation process (FFPE) of tumour tissues (Costello et al., 2013). Indels were called with Strelka (v1.0.11). MuTect calls and Strelka calls were further filtered through a panel of normal samples (PoN) to remove artifacts generated by rare error modes and miscalled germline |

alterations 65. To pass quality control, samples were required to have <5% cross-sample contamination as assessed with ContEst (Cibulskis et al., 2013); mean target coverage of at least 25x in the tumour sample and 20x in the corresponding normal as assessed using GATK3.7 DepthOfCoverage; and a percentage of tumour-in-normal of < 30% as determined by deTiN 68. This pipeline was modified for analysis of cell lines rather than tumour-normal pairs as follows: indels were called through MuTect2 alone rather than Strelka; deTiN was not performed; and a common variant filter was applied to exclude variants present in The Exome Aggregation Consortium (ExAC) if at least 10 alleles containing the variant were present across any subpopulation, unless they appeared in a list of known somatic sites (Lek et al. 2016 and AACR Project Genei Consortium, 2017).

Mutational signature analysis:
Active mutational processes (Alexandrov et al., 2013) were determined using the deconstructSigs R package, with a signature contribution cutoff of 6%. This cutoff was chosen because it was the minimum contribution value required to obtain a false-positive rate of 0.1% and false-negative rate of 1.4% per the authors' in-silico analysis, and is the recommended cutoff (Rosenthal et al., 2016). Samples with < 10 mutations were excluded from analysis due to poor signature discrimination with so few mutations.

RNA-Seq analyses: PDX-tissue RNA extractions were carried using RNeasy micro kit (Qiagen). RNA-Seq was performed using replicate samples on the Illumina HiSeq4000, paired-end 100-bp reads at the Center for Advanced Technology (UCSF). For the differential gene expression analysis, DESeq program was used to compare controls to erlotinib samples as previously described (Anders & Huber, 2010).
RNAseq samples form patients and cell lines were sequenced by Novogene (https://en.novogene.com/) with paired-end sequencing (150bp in length). There were ~20 million reads for each sample. The processed fastq files were mapped to hg19 reference genome using STAR (version 2.4) algorithm and transcript expressions were quantified using RSEM (version 1.2.29) algorithm. The default parameters in the algorithms were used. The normalized transcript reads (TPM) were used for downstream analysis.
For single cell RNA Seq analyses the data from a previously published study (excluding samples from the neoadjuvant osimertinib dataset) was used and analyzed in a similar manner (Maynard et al., 2020). All cells used are identified as malignant by marker expression and CNV inference and originated in from various biopsy sites (adrenal, liver, lymph node, lung, pleura/pleural fluid). Nonparametric, pairwise comparisons (Wilcoxon Rank Sum Test) was used to determine the statistical significance of the pairwise comparisons of different timepoints for their average scaled expression.

Human EGFR transgene amplicon sequencing of mouse: FASTQ files were aligned to hg19 obtained from the GATK bundle (v2.8) using bwa mem (bwa v0.7.15)98,99. Analyses were performed using R (version 3.3.1) and deepSNV (v1.18.1)100. The median depth of coverage of sequenced EGFR exons (19,20,21) was 5290x (range: 2238-8040). Variants associated with resistance to EGFR tyrosine kinase inhibitors were queried using deepSNV's bam2R function, with the arguments q=20 & s=2. The variants explored include: T790M, D761Y, L861Q, G796X, G797X, L792X, L747S. L858R was identified in every sequenced sample.

Whole exome sequencing – mouse data: WES was performed by the Advanced Sequencing Facility at The Francis Crick Institute using the Twist BioScience Human Core Exon Kit for library preparation and Agilent SureSelectXT Mouse All Exon, 16, Kit for library preparation respectively. Sequencing was performed on HiSeq 4000 platforms.
RNA sequencing – mouse data: RNA-seq was performed by the Advanced Sequencing Facility at the Francis Crick Institute using the KAPA mRNA HyperPrep Kit (KK8581 – 96 Libraries) and KAPA Dual-Indexed Adapters (Roche-KK8720). Sequencing was performed on HiSeq 4000 platforms. The processed fastq files were mapped to mm10 reference genome using STAR (version 2.4) algorithm and transcript expressions were quantified using RSEM (version 1.2.29) algorithm with the default parameters. The read counts were used for downstream analysis.
Alignment – mouse: All samples were de-multiplexed and the resultant FASTQ files aligned to the mm10 mouse genome, using bwa mem (bwa v0.7.15). De-duplication was performed using Picard (v2.1.1) (http://broadinstitute.github.io/picard). Quality control metrics were collated using FASTQC (v0.10.1 - http://www.bioinformatics.babraham.ac.uk/projects/fastqc/), Picard and GATK (v3.6). SAMtools (v1.3.1) was used to generate mpileup files from the resultant BAM files. Thresholds for base phred score and mapping quality were set at 20. A threshold of 50 was set for the coefficient of downgrading mapping quality, with the argument for base alignment quality calculation being deactivated. The median depth of coverage for all samples was 92x (range: 58-169x).
Variant detection & annotation – mouse: Variant calling was performed using VarScan2(v2.4.1), MuTect(v1.1.7) and Scalpel(v0.5.4)91-93. The following argument settings were used for variant detection using VarScan2:
--min-coverage 8 --min-coverage-normal 10 --min-coverage-tumor 6 --min-var-freq 0.01 --min-freq-for-hom 0.75 --normal-purity 1 --p-value 0.99 --somatic-p-value 0.05 --tumor-purity 0.5 --strand-filter 0
For MuTect, only "PASS" variants were used for further analyses. With the exception of allowing variants to be detected down to a VAF of 0.001, default settings were used for Scalpel insertion/deletion detection.
To minimise false positives, additional filtering was performed. For single-nucleotide variants (SNVs) or dinucleotides detected by VarScan2, a minimum tumor sequencing depth of 30, variant allele frequency (VAF) of 5%, variant read count of 5, and a somatic p-value <0.01 were required to pass a variant. For variants detected by VarScan2 between 2 and 5% VAF, the mutation also needs to be detected by MuTect.
As for insertions/deletions (INDELs), variants need to be passed by both Scalpel ("PASS") and VarScan2 (somatic p value <0.001). A minimum depth of 50x, 10 alt reads and VAF of 2% was required.
For all SNVs, INDELs and dinucleotides, any variant also detected in the paired germline sample with more than 5 alternative reads or a VAF greater than 1% was filtered out.
The detected variants were annotated using Annovar94.
Functional annotation of SNVs – mouse: Murine gene mutation callings from whole exome sequencing were parsed with some modification including genomic coordinates (removing 'chr' before chromosomal numbers, only 'SNV' was selected). The modified files were fed into PROVEAN (Protein Variation Effect Analyzer)95-97 software tool (http://provean.jcvi.org/index.php) to predict whether an amino acid substitution has an impact on the biological function of a protein (SIFT score). The predict files were merged with original files at gene level annotation using R program.

For manuscripts utilizing custom algorithms or software that are central to the research but not yet described in published literature, software must be made available to editors and reviewers. We strongly encourage code deposition in a community repository (e.g. GitHub). See the Nature Portfolio guidelines for submitting code & software for further information.

# Data

Policy information about availability of data

All manuscripts must include a data availability statement. This statement should provide the following information, where applicable:
- Accession codes, unique identifiers, or web links for publicly available datasets
- A description of any restrictions on data availability
- For clinical datasets or third party data, please ensure that the statement adheres to our policy

The Whole Exome Sequencing (WES) data and RNAseq data (from the TRACERx study) generated, used or analysed during this study are available through the Cancer Research UK and University College London Cancer Trials Centre (ctc.tracerx@ucl.ac.uk) for academic non-commercial research purposes only, and subject to review of a project proposal that will be evaluated by a TRACERx data access committee, entering into an appropriate data access agreement and subject to any applicable ethical approvals. The Whole Genome Sequencing (WGS) data shown in Fig. 6 will be available upon request, subject to evaluation by corresponding authors. For the single cell RNA-Seq analyses shown in Ext. Data Fig. 10b-c, the data from a previously published study (all advanced lung cancer cell data) were used and analyzed in a similar manner31. This data is available as an NCBI Bioproject # PRJNA591860. The RNA-Seq data for Ext. Data Fig. 10a was from a previously published study1. These data are available at NCBI GEO under accession number GSE65420. Related RNA-Seq and WES sequencing data will be deposited into NCBI GEO and SRA database.

# Research involving human participants, their data, or biological material

Policy information about studies with human participants or human data. See also policy information about sex, gender (identity/presentation), and sexual orientation and race, ethnicity and racism.

| | |
|---|---|
| Reporting on sex and gender | Participant sex (biological attribute) provided in supplementary tables. No sex- and gender-based analysis was performed, not relevant and outside of scope of current study. |
| Reporting on race, ethnicity, or other socially relevant groupings | Participant race provided in supplementary tables. No race-based analysis was performed, not relevant and outside of scope of current study. |
| Population characteristics | Population characteristics relevant in this study: <br> - Age (adult over 18) <br> - Lung cancer histology <br> - Oncogenic driver mutation <br> - Treatment history <br> - Treatment response <br><br> Other population characteristics provided but not relevant in this study: <br> - Smoking history |
| Recruitment | Patients were recruited according to Institutional Review Board-approved protocols CC13-6512 and CC17-658, NCT03433469. No bias to report. |
| Ethics oversight | All patients gave informed consent for collection of clinical correlates, tissue collection, research testing under Institutional Review Board-approved protocols listed above. Patient demographics are listed in Supplemental Tables. Patient studies were conducted according to the Declaration of Helsinki, the Belmont Report, and the U.S. Common Rule. |

Note that full information on the approval of the study protocol must also be provided in the manuscript.

# Field-specific reporting

Please select the one below that is the best fit for your research. If you are not sure, read the appropriate sections before making your selection.

☒ Life sciences  ☐ Behavioural & social sciences  ☐ Ecological, evolutionary & environmental sciences

For a reference copy of the document with all sections, see nature.com/documents/nr-reporting-summary-flat.pdf

# Life sciences study design

All studies must disclose on these points even when the disclosure is negative.

| | |
|---|---|
| Sample size | Sample size of clinical data was determined by access and analysis of all available cases that met criteria. Sample size of mouse experiments were determined based on previous work done using these same models (E.C. de Bruin et al. 2014), and based on previous work in the laboratory. |
| Data exclusions | Details of exclusion criteria based are described in detail in the methods section. |
| Replication | Studies involving pre-clinical models were performed with two or more biological and/or technical replicates. |

| Randomization | Mice were assigned to vehicle or treatment groups for xenograft studies to ensure even distribution in both groups based on the tumor size just prior to the initiation of treatment. Genetically modified mice were placed in groups based on their genetic background. |
|---|---|
| Blinding | Investigators were blinded during data collection, analysis of mouse tissues, and microCT analysis. |

# Reporting for specific materials, systems and methods

We require information from authors about some types of materials, experimental systems and methods used in many studies. Here, indicate whether each material, system or method listed is relevant to your study. If you are not sure if a list item applies to your research, read the appropriate section before selecting a response.

### Materials & experimental systems

| n/a | Involved in the study |
|---|---|
| ☐ | ☒ Antibodies |
| ☐ | ☒ Eukaryotic cell lines |
| ☒ | ☐ Palaeontology and archaeology |
| ☐ | ☒ Animals and other organisms |
| ☐ | ☒ Clinical data |
| ☒ | ☐ Dual use research of concern |
| ☒ | ☐ Plants |

### Methods

| n/a | Involved in the study |
|---|---|
| ☒ | ☐ ChIP-seq |
| ☒ | ☐ Flow cytometry |
| ☒ | ☐ MRI-based neuroimaging |

## Antibodies

| Antibodies used | Antibodies used for western Blot analysis: pEGFR (CST-3777 or 2236), pERK (CST-4370 or 9106), APOBEC3B (A gift from Harris Lab, Brown et al., 2019), UNG (A gift from Harris Lab, Serebrenik et al., 2019), GAPDH (sc-59540), Histone H3 (CST-9715), EGFR (CST-4267), TUBB (CST-2146), Hsp90 (CST-4874), RELA (CST-8242), RELB (CST- 4922),ERK (CST- 9102), pSTAT3 (CST-9145), STAT3 (CST-9139), AKT (CST-2920) and pAKT (CST-4060).<br><br>Antibodies used for IHC analysis:<br>EGFRL858R mutant specific (Cell Signaling: 3197, 43B2)<br>APOBEC3B (5210-87-13, Brown et al,.2019)<br>Ki67 (Abcam: Ab15580)<br>Caspase 3 (R&D (Bio-Techne): AF835)<br>p-Histone H2AX (Sigma-Aldrich, 05-636),<br>UNG (NB600-1031, Novus Biologicals)<br>p53 (Leica, cat # NCL-L-p53-CM5p)<br>phospho-HIstone H3 (Sigma-Aldrich, Ser10) |
|---|---|
| Validation | Validation related to western blot analysis: APOBEC3B, UNG antibody was validated with RNAi and/or CRISPR-Cas9 mediated approaches with orthogonal validation using RT-qPCR approach. All other antibodies were validated with RNAi mediated knockdown approaches or based on the expected changes in those proteins upon treatment with inhibitors. Most of the antibodies used in this study have been extensively used in previously published studies from our laboratory (Blakely et al., 2015, Hrustanovic et al., 2015).<br><br>Validation related to IHC analysis: All of the antibodies, except p53 were previously optimized and validate in the Experimental Histopathology Unit (EHP) at the Francis Crick Institute. p53 has been validated extensively in the Vousden  Laboratory and Attardi Laboratories, and by Leica Biosystems.<br><br>APOBEC3B and UNG were optimized using mouse tissue and cell lines that served as positive and negative controls. The optimization of APOBEC3B was based off of the publication Brown et al., 2019. The optimization of UNG was done using information from Novus Biologicals (https://www.novusbio.com/products/ung-antibody_nb600-1031#datasheet) and from previous publications using this antibody (Yang et al., 2005.) |

## Eukaryotic cell lines

Policy information about cell lines and Sex and Gender in Research

| Cell line source(s) | Cell lines were purchased from ATCC as described in the previous studies from our laboratory (Blakely et al., 2015, Hrustanovic et al., 2015). Mouse tumour cell lines were generated from tumours of genetically engineered mouse models mentioned in the manuscript, and verified by flow cytometry analysis and gene expression analysis. |
|---|---|
| Authentication | Cell lines were previously validated by STR analysis. |
| Mycoplasma contamination | Cell lines were tested for mycoplasma contamination. |
| Commonly misidentified lines (See ICLAC register) | N/A |

# Animals and other research organisms

Policy information about studies involving animals; ARRIVE guidelines recommended for reporting animal research, and Sex and Gender in Research

| | |
|---|---|
| Laboratory animals | For xenograft studies 6-8 week old female NOD/SCID mice were used.<br><br>For GEM model studies:<br>Briefly, all mice were purified to the C57BL/6J through back crossing of at least 8 generations or purity was assessed using the a C57BL/6J substrain panel.<br><br>Genotypes of mice used in the study:<br>- TetO-EGFRL858R;R26tTA/+ (E)<br>- TetO-EGFRL858R;R26tTA/LSL-APOBEC3B (EA3B)<br>- TetO-EGFRL858R;CCSP-rtTA:R26LSL-APOBEC3B/Ert2-Cre (EA3Bi)<br><br>An approximately equal number of males and females were used in each arm of the study. Mice were housed with up to 5 mice per cage and separated by sex. Feed and water were available ad libitum.<br><br>Diet information:<br>- Regular diet: 2018s Global Rodent 18% autoclavable diet<br>- Doxycycline diet (625 ppm (Harlan-Tekland) Irradiated diet)<br><br>Water information: Mice have free access to RO water supplied by automatic watering system.<br><br>Caging information: Techniplast green line IVC cages<br><br>Chamber environment information: Mice are housed on aspen wood chip, and each cage is provided with nesting material, a mouse house and mouse loft for environmental enrichment.<br><br>Type and frequency of observation: Mice were checked once daily. Animals in each experiment were weighed initially and then weekly for the duration of the experiment. Mice were<br><br>Humane endpoints (clinical signs or situations that determine the end of the experiment for any animal under this experiment to prevent unnecessary suffering):<br><br>Mice which showed any one of the signs below were culled by an overdose of anaesthetic and then a terminal bleed (exsanguination) was performed.<br><br>If weight loss reaches 10% of original body weight within 24 hours, or 15% weight loss within 48 hours, or 15-20% over a longer period<br>• Persistent hunched posture/piloerection and laboured breathing<br>• Diarrhoea<br>• If ulceration or infection are noted<br>• If persistent self-induced trauma is noted |
| Wild animals | N/A |
| Reporting on sex | For genetically engineered mouse models, both male and female animals were used equally. For xenograft mouse models, only female NOD/SCID mice were used according to previously established UCSF IACUC-approved animal protocols. No sex-based analysis was performed, not relevant and outside of scope of current study. |
| Field-collected samples | N/A |
| Ethics oversight | The xenograft studies were approved and overseen by the UCSF IRB. All GEM model animal regulated procedures were approved by the Francis Crick Institute BRF Strategic Oversight Committee that incorporates the Animal Welfare and Ethical Review Body and conformed with the UK Home Office guidelines and regulations under the Animals (Scientific Procedures) Act 1986 including Amendment Regulations 2012. All xenograft models were conducted under UCSF IACUC-approved animal protocols. |

Note that full information on the approval of the study protocol must also be provided in the manuscript.

# Clinical data

Policy information about clinical studies

All manuscripts should comply with the ICMJE guidelines for publication of clinical research and a completed CONSORT checklist must be included with all submissions.

| | |
|---|---|
| Clinical trial registration | NCT03433469 |
| Study protocol | https://clinicaltrials.gov/study/NCT03433469#study-plan |
| Data collection | Patients were recruited for this study from July 1 2018 to August 1, 2020. Data were collected for this study at UCSF from July 1, |

Data collection | 2018 to December 1, 2020.

Outcomes | The primary outcome of this study is to evaluate the efficacy of osimertinib as neoadjuvant therapy in patients with surgically resectable EGFR-mutant NSCLC. The primary endpoint of the study will be MPR rate defined as ≤10% viable tumor present histologically in the resected tumor specimen. Secondary measures of efficacy are radiographic decrease in maximum tumor diameter, 5-year DFS, 5-year OS, pathological response rate (pCR), and depth of response (DpR). Exploratory endpoints are to evaluate genomic and transcriptional changes on baseline and resected tumor specimens as determined by whole exome sequencing and RNA sequencing.

