## [Peer Review File · Nature Genetics]

Peer Review Information

Manuscript Title: The role of APOBEC3B in lung tumor evolution and targeted cancer therapy resistance

Corresponding author name(s): Professor Trever (G) Bivona, Dr Deborah (R.) Caswell

Editorial Notes:

Transferred manuscripts This document only contains reviewer comments, rebuttal and decision letters for versions considered at Nature Genetics.

Reviewer Comments & Decisions:

Decision Letter, initial version:
--

14th Aug 2023

Dear Dr Bivona,

Thank you for submitting your revised manuscript "The role of APOBEC3B in lung tumor evolution and targeted therapy resistance" (NG-A62839-T). It has now been seen by the original referees and their comments are below. The reviewers find that the paper has improved in revision, and therefore we'll be happy in principle to publish it in Nature Genetics, pending minor revisions to satisfy the referees' final requests and to comply with our editorial and formatting guidelines.

As discussed by email last week, we would strongly encourage you to use transparent peer-review, which will allow us to publish the latest round of peer-review and decision letters with your readers. As you know, this option precludes publication of peer-review reports that were obtained when the paper was being considered elsewhere. As such, we'd like you to prepare a Supplementary note (that is referred to in the main manuscript) that discusses all the main points raised during all rounds of peer-review. This Supplementary note will be subject to editorial review. Our aim is to provide a balanced and scholarly discussion of the work in the context of some of the controversies within the field (particularly those which were raised by your Reviewers), and the limits of your experimental approach.

Sincerely,

Safia Danovi
Editor
Nature Genetics

Reviewer #4 (Remarks to the Author):

I perused the authors' responses to the latest set of comments for the manuscript "The role of APOBEC3B in lung tumor evolution and targeted therapy resistance" by Caswell et al. I was also informed by the editorial staff that, from a novelty perspective, Nature Genetics is comfortable with the somewhat correlative nature of this study provided that the study's limitations are clearly stated in the text.

As the authors have not performed the experiments suggested in the prior set of reviews, my general concerns remain unchanged. Briefly, I do not find that the current study provides sufficient evidence demonstrating that APOBEC3B plays a causative role in lung tumor evolution and/or targeted therapy resistance. In my opinion, while APOBEC3B may indeed play such a role, the presented results are generally subject to different biases (as outlined by Referee #2) or mostly correlative. The authors have modified the text and, indeed, their statements are more consistent with the results from the study and the changes outline some of the limitations. Nevertheless, the changes in the manuscript do not fully reflect the concerns raised by me or the other reviewer and, after the inevitable shortening of the text to fit the paper within the word limits of the journal, I suspect that these limitations will be missed by most readers.

As such, I am only comfortable recommending the manuscript for publication provided that the peer review file is included with publication. This will provide a valuable resource of the disagreements between authors and reviewers, outline all limitations listed during the review process, as well as clearly show that, even though the manuscript has been under review for several years in Nature, the authors did not perform multiple requested experiments and tend to favored a response of "Unfortunately, due to time constraints we were unable to perform this experiment."

Final Decision Letter:

25th Oct 2023

Dear Dr Bivona,

I am delighted to say that your manuscript "The role of APOBEC3B in lung tumor evolution and targeted cancer therapy resistance" has been accepted for publication in an upcoming issue of Nature Genetics.

Your paper will be published online after we receive your corrections and will appear in print in the next available issue. You can find out your date of online publication by contacting the Nature Press Office (press@nature.com) after sending your e-proof corrections. Now is the time to inform your Public Relations or Press Office about your paper, as they might be interested in promoting its publication. This will allow them time to prepare an accurate and satisfactory press release. Include your manuscript tracking number (NG-A62839R) and the name of the journal, which they will need when they contact our Press Office.

Please note that *Nature Genetics* is a Transformative Journal (TJ). Authors may publish their research with us through the traditional subscription access route or make their paper immediately open access through payment of an article-processing charge (APC). Authors will not be required to make a final decision about access to their article until it has been accepted. [Find out more about Transformative Journals](https://www.springernature.com/gp/open-research/transformative-journals)

Authors may need to take specific actions to achieve  > **compliance with funder and institutional open access mandates**. If your research is supported by a funder that requires immediate open access (e.g. according to [Plan S principles](https://www.springernature.com/gp/open-research/plan-s-compliance)) then you should select the gold OA route, and we will direct you to the compliant route where possible. For authors selecting the subscription publication route, the journal's standard licensing terms will need to be accepted, including <https://www.nature.com/nature-portfolio/editorial-policies/self-archiving-and-license-to-publish>. Those licensing terms will supersede any other terms that the author or any third party may assert apply to any version of the manuscript.

If you have not already done so, we invite you to upload the step-by-step protocols used in this manuscript to the Protocols Exchange, part of our on-line web resource, natureprotocols.com. If you complete the upload by the time you receive your manuscript proofs, we can insert links in your article that lead directly to the protocol details. Your protocol will be made freely available upon publication of your paper. By participating in natureprotocols.com, you are enabling researchers to more readily reproduce or adapt the methodology you use. [Natureprotocols.com](http://natureprotocols.com) is fully searchable, providing your protocols and paper with increased utility and visibility. Please submit your protocol to <https://protocolexchange.researchsquare.com/>. After entering your [nature.com](http://www.nature.com) username and password you will need to enter your manuscript number (NG-A62839R). Further information can be found at <https://www.nature.com/nature-portfolio/editorial-policies/reporting-standards#protocols>

Sincerely,

Safia Danovi
Editor
Nature Genetics